# Deep Backtracking Counterfactuals for Causally Compliant Explanations

**Klaus-Rudolf Kladny**                                         *kkladny@tue.mpg.de*
*Max Planck Institute for Intelligent Systems, Tübingen, Germany*

**Julius von Kügelgen**                        *julius.vonkuegelgen@stat.math.ethz.ch*
*ETH Zurich, Switzerland*

**Bernhard Schölkopf**                                              *bs@tue.mpg.de*
*Max Planck Institute for Intelligent Systems, Tübingen, Germany*

**Michael Muehlebach**                                      *michaelm@tue.mpg.de*
*Max Planck Institute for Intelligent Systems, Tübingen, Germany*

**Reviewed on OpenReview:** *https://openreview.net/forum?id=Br5esc2CXR*

## Abstract

Counterfactuals answer questions of what would have been observed under altered circumstances and can therefore offer valuable insights. Whereas the classical interventional interpretation of counterfactuals has been studied extensively, *backtracking* constitutes a less studied alternative where all causal laws are kept intact. In the present work, we introduce a practical method called *deep backtracking counterfactuals* (DeepBC) for computing backtracking counterfactuals in structural causal models that consist of deep generative components. We propose two distinct versions of our method—one utilizing Langevin Monte Carlo sampling and the other employing constrained optimization—to generate counterfactuals for high-dimensional data. As a special case, our formulation reduces to methods in the field of counterfactual explanations. Compared to these, our approach represents a causally compliant, versatile and modular alternative. We demonstrate these properties experimentally on a modified version of MNIST and CelebA.

## 1 Introduction

In recent years, there has been a surge in the use of deep learning for causal modeling (Sanchez & Tsaftaris, 2022; Pawlowski et al., 2020; Kocaoglu et al., 2018; Goudet et al., 2018; Javaloy et al., 2023; Khemakhem et al., 2021; Taylor-Melanson et al., 2024). The integration of deep learning in causal modeling combines the potential to effectively operate on high-dimensional distributions, a strength inherent to deep neural networks, with the capability to answer inquiries of a causal nature, thus going beyond statistical associations. At the apex of such inquiries lies the ability to generate scenarios of a counterfactual nature—altered worlds where variables differ from their factual realizations, hence aptly termed *counter to fact* (Pearl, 2009; Bareinboim et al., 2022). Counterfactuals are deeply ingrained in human reasoning (Roese, 1997), as evident from phrases such as *"Had it rained, the grass would be greener now"* or *"Had I invested in bitcoin, I would have become rich"*.

Constructing counterfactuals necessitates two fundamental components: (i) a sufficiently accurate world model with mechanistic semantics, such as a structural causal model (SCM; Pearl, 2009); and (ii) a sound procedure for deriving the distribution of all variables that are not subject to explicit alteration. The latter component has been a subject of debate: While the classical literature in causality constructs counterfactuals by actively manipulating causal relationships (*interventional counterfactuals*), this approach has been contested by some psychologists and philosophers (Rips, 2010; Gerstenberg et al., 2013; Lucas & Kemp, 2015).

Figure 1: **Visualization of DeepBC for Morpho-MNIST.** We generate a counterfactual (green) image img$^*$ and thickness $t^*$ with antecedent intensity $i^*$ for the factual, observable realizations (filled blue) img, $t$, $i$. Our approach finds new latent variables $\mathbf{u}^*$ that are close with respect to distances $d_i$ to the factual latents $\mathbf{u}$, subject to rendering the antecedent $i^*$ true. The causal mechanisms in the factual world remain unaltered in the counterfactual world. In this specific distribution, thickness and intensity are positively related, thus rendering the image both more intense and thicker in the counterfactual. Dependence of $f_i$ on graphical parents is omitted for simplifying visual appearance.

Instead, they have proposed an account of counterfactuals where alternate worlds are derived by tracing changes back to background conditions while leaving all causal mechanisms intact. This type of counterfactual is termed *backtracking counterfactual* (Lewis, 1979; Khoo, 2017). Due to the preservation of causal mechanisms, backtracking counterfactuals allow for gaining faithful insights into the structural relationships of the data generating process, which render them a promising opportunity in practical domains such as medical imaging (Sudlow et al., 2015), biology (Yang et al., 2021a) and robotics (Ahmed et al., 2021).

Recently, von Kügelgen et al. (2023b) have formalized backtracking counterfactuals within the SCM framework. However, implementing this formalization for deep structural causal models (Pawlowski et al., 2020) poses computational challenges due to steps such as marginalizations and the evaluation of distributions that are intractable. The present work addresses these challenges and offers a computationally efficient implementation by framing the generation of counterfactuals as a constrained sampling problem. Specifically, we propose a Markov Chain Monte Carlo scheme in the structured latent space of a causal model, based on the overdamped Langevin dynamics (Parisi, 1981). We also propose a simplified method where a single, "most likely" counterfactual is obtained as the solution of a constrained optimization problem.

The present work further serves as a bridge between causal modeling and practical methods in the field of counterfactual explanations (Wachter et al., 2017; Beckers, 2022). As a causally grounded approach applicable to high-dimensional data, our method fills a gap in the existing literature between non-causal explanation tools, built for complex data such as images (e.g., Goyal et al., 2019; Boreiko et al., 2022), and causal methods that have only been applied to simple (assuming additive noise), low-dimensional settings (Bynum et al., 2024; von Kügelgen et al., 2023b; Crupi et al., 2022).

We summarize our main contributions as follows:

- We introduce DeepBC, a tractable method for computing backtracking counterfactuals in deep SCMs (§ 3). We propose two variants, *stochastic DeepBC* (§ 3.1.1) and *mode DeepBC* (§ 3.1.2). The former

allows for sampling counterfactuals via Langevin Monte Carlo (§ 3.3.1). The latter constitutes a simplified version for generating point estimates using constrained optimization (§ 3.3.2). Our methodology exhibits multiple favorable properties, which are causal compliance, versatility and modularity (§ 3.5).

- We highlight connections to the field of counterfactual explanations, and elucidate how our method can be understood as a general form of the popular method proposed by Wachter et al. (2017) (§ 3.2).

- We demonstrate the applicability and distinct advantages of our method in comparison to interventional counterfactuals and counterfactual explanation methods through experiments on the Morpho-MNIST and the CelebA data sets (§ 4).

**Overview.** Section § 2 introduces structural causal models (§ 2.1), the deep generative models that are employed subsequently (§ 2.2), interventional and backtracking counterfactuals (§ 2.3) and counterfactual explanations (§ 2.4). The section therefore sets the stage for presenting our method called *deep backtracking counterfactuals* (DeepBC) in Section § 3, where we also discuss its relation to methods in the field of counterfactual explanations (§ 3.2), its algorithmic implementation (§ 3.3) and extensions for categorical variables and sparse solutions (§ 3.4). In Section § 4, we show experimental results performed on Morpho-MNIST (§ 4.1) and CelebA (§ 4.2) that highlight the causal compliance, versatility and modularity of our method. Related work is presented in Section § 5. We then discuss limitations and future work in Section § 6 and conclude with a short summary in Section § 7.

## 2 Setting & Preliminaries

The following section introduces (deep) structural causal models and backtracking counterfactuals. These concepts present the building blocks for our method, presented in Section § 3.

Throughout the article, upper case $X$ denotes a scalar or multivariate continuous random variable, and lower case $x$ a realization thereof. Bold $\mathbf{X}$ denotes a collection of such random variables with realizations $\mathbf{x}$. The components of $\mathbf{x}$ are denoted by $x_i$. We denote the probability density of $X$ by $p(x)$.

### 2.1 Structural Causal Models

Let $\mathbf{X} = (X_1, X_2, ..., X_n)$ be a collection of potentially high-dimensional observable "endogenous" random variables. For instance, these variables could be high-dimensional objects such as images (e.g., the MNIST image in Fig. 1) or scalar feature variables (such as $t$ and $i$ in Fig. 1). The causal relationships among the $X_i$ are specified by a directed acyclic graph $G$ that is known. A structural causal model (Pearl, 2009) is characterized by a collection of structural equations $X_i \leftarrow f_i(\mathbf{X}_{\mathrm{pa}(i)}, U_i)$, for $i = 1, 2, ..., n$, where $\mathbf{X}_{\mathrm{pa}(i)}$ are the causal parents of $X_i$ as specified by $G$ and $\mathbf{U} = (U_1, U_2, ..., U_n)$ are exogenous latent variables[1]. The acyclicity of $G$ ensures that for all $i$, we can recursively solve for $X_i$ to obtain a deterministic expression in terms of $\mathbf{U}$. Thus, there exists a unique function that maps $\mathbf{U}$ to $\mathbf{X}$, which we denote by $\mathbf{F}$,

$$\mathbf{X} = \mathbf{F}(\mathbf{U}), \tag{1}$$

and which is known as the reduced-form expression. We see that $\mathbf{F}$ induces a distribution over observables $\mathbf{X}$, for any given distribution over the latents $\mathbf{U}$. For the remainder of this work, we assume causal sufficiency (no unobserved confounders) (Spirtes, 2010), which implies joint independence of the components of $\mathbf{U}$.

### 2.2 Deep Invertible Structural Causal Models

In this work, we make the simplifying assumption that $f_i(\mathbf{x}_{\mathrm{pa}(i)}, \cdot)$ is invertible for any fixed $\mathbf{x}_{\mathrm{pa}(i)}$[2] such that we can write

$$U_i = f_i^{-1}(\mathbf{X}_{\mathrm{pa}(i)}, X_i), \quad i = 1, 2, ..., n.$$

---

[1]We note that by definition, the exogenous variables are not causally related to each other.
[2]also known as *bijective generation mechanism* (see, e.g., Nasr-Esfahany et al., 2023).

Under this assumption, the inverse $\mathbf{F}^{-1}$ of the mapping in (1) is guaranteed to exist, and we can write

$$\mathbf{U} = \mathbf{F}^{-1}(\mathbf{X}). \tag{2}$$

We assume that all $f_i$ are given as conditional deep generative models, differentiable with respect to $u_i$ for each $\mathbf{x}_{\mathrm{pa}(i)}$, trained separately for each structural assignment (Pawlowski et al., 2020). We consider the following two classes of models, both of which operate on latent variables with a standard Gaussian prior.

**Conditional normalizing flows** (Rezende & Mohamed, 2015; Winkler et al., 2019) are constructed as a composition of invertible functions, hence rendering the entire function $f_i$ invertible in $u_i$. In addition, they are chosen such that the determinant of the Jacobian can be computed efficiently. These two attributes facilitate efficient training of $f_i$ via maximum likelihood.

**Conditional variational auto-encoders** (Kingma & Welling, 2014; Sohn et al., 2015) consist of separate encoder $e_i$ and decoder $d_i$ networks. These modules parameterize the mean of their respective conditional distributions, i.e., $U_i|\mathbf{x}_{\mathrm{pa}(i)}, x_i \sim \mathcal{N}(e_i(\mathbf{x}_{\mathrm{pa}(i)}, x_i), \mathrm{diag}(\boldsymbol{\sigma}_e^2))$ and $X_i|\mathbf{x}_{\mathrm{pa}(i)}, u_i \sim \mathcal{N}(d_i(\mathbf{x}_{\mathrm{pa}(i)}, u_i), \mathbf{I}\sigma_d^2)$. Through joint training of $e_i$, $d_i$ and the variance vector $\boldsymbol{\sigma}_e^2$ using variational inference, $e_i$ and $d_i$ become interconnected. The decoder variance $\sigma_d^2$ is fixed a priori. Theoretical insights by Reizinger et al. (2022) support the use of an approximation, where the decoder effectively inverts the encoder, that is,

$$x_i = f_i(\mathbf{x}_{\mathrm{pa}(i)}, f_i^{-1}(\mathbf{x}_{\mathrm{pa}(i)}, x_i)) \approx d_i(\mathbf{x}_{\mathrm{pa}(i)}, e_i(\mathbf{x}_{\mathrm{pa}(i)}, x_i)).$$

We use this approximation throughout the present work and do not explicitly model the encoder and decoder variances post training. The reason is that invertability is crucial to the simplification of the backtracking procedure, as derived in App. A.2.

## 2.3 Interventional and Backtracking Counterfactuals

Given a factual observation $\mathbf{x}$ (blue in Fig. 2) and a so-called antecedent $\mathbf{x}_S^* = (x_i^* : i \in S)$ (filled green in Fig. 2) for a given subset $S \subset \{1, 2, ...., n\}$, we define a counterfactual as some $\mathbf{x}^* = (x_1^*, x_2^*, ..., x_n^*)$ consistent with $\mathbf{x}_S^*$. We view $\mathbf{x}^*$ as an answer to the verbal query

> "What values ($\mathbf{x}^*$) had $\mathbf{X}$ taken instead of the given (observed) $\mathbf{x}$, had $\mathbf{X}_S$ taken the values $\mathbf{x}_S^*$ rather than $\mathbf{x}_S$?"

In the present work, we consider interventional and backtracking counterfactuals. Both generate distributions over counterfactuals whose random variables we refer to as $\mathbf{X}^*$ (green in Fig. 2). We only provide a conceptual notion and refer the reader to App. A.1 for a more rigorous formalism for both types of counterfactuals.

**Interventional counterfactuals** render the antecedent true via modification of the structural assignments $(f_1, f_2, ..., f_n)$, which leads to a new collection of assignments $(f_1^*, f_2^*, ..., f_n^*)$. Specifically, these new structural assignments are constructed such that the causal dependence on the causal parents of all antecedent variables $\mathbf{X}_S^*$ is removed: $f_i^* = x_i^*$ for $i \in S$ and $f_i^* = f_i$ otherwise. Such a modification can be understood as a *hard intervention* on the underlying structural relations (indicated by the hammer in Fig. 2).

**Backtracking counterfactuals** leave all structural assignments unchanged. In order to set the antecedent $\mathbf{x}_S^* \neq \mathbf{x}_S$ true, they trace differences to the factual realization back to (ideally small) changes in the latent variables $\mathbf{U}$. These modified latent variables are represented by a new collection of variables $\mathbf{U}^*$ that depend on $\mathbf{U}$ via a given backtracking conditional $p^B(\mathbf{u}' | \mathbf{u})$ (red in Fig. 2) (von Kügelgen et al., 2023b), which represents a probability density for computing similarity between $\mathbf{u}$ and $\mathbf{u}'$ and which we assume to be decomposable, or factorized:

$$p^B(\mathbf{u}' | \mathbf{u}) = \prod_{i=1}^{n} p_i^B(u_i' | u_i).$$

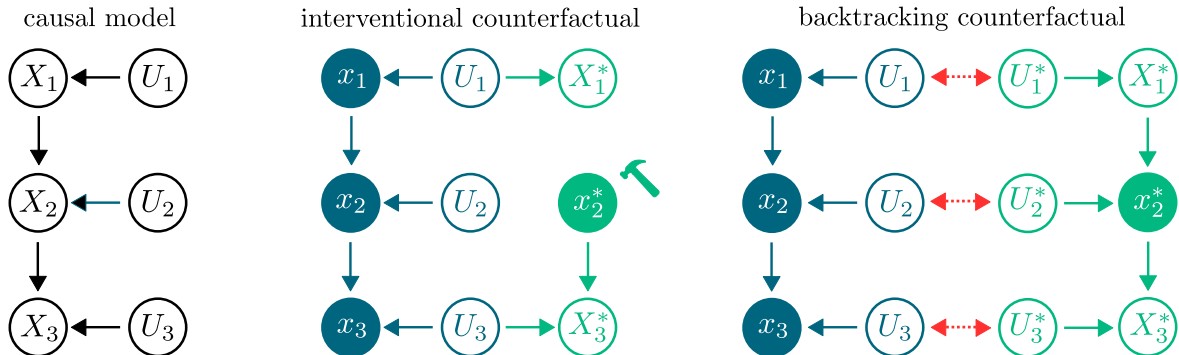

Figure 2: **Difference between interventional and backtracking counterfactuals on a concrete example.** Variables that are conditioned on correspond to filled circles. Interventional counterfactuals perform a hard intervention (indicated by a hammer) $X_2^* \leftarrow x_2^*$ with antecedent $x_2^*$ (i.e., $S = \{2\}$) in the counterfactual world (green). Backtracking counterfactuals, on the contrary, construct this counterfactual world via introducing a new set of latent variables $\mathbf{U}^*$ that depend on $\mathbf{U}$ via a backtracking conditional (red).

By marginalizing over $\mathbf{U}^*$, we obtain the distribution of $\mathbf{X}^* \,|\, \mathbf{x}_S^*, \mathbf{x}$. Throughout the manuscript, we will write

$$p_i^B(u_i' \,|\, u_i) \;\propto\; \exp\left\{ -d_i(u_i', u_i) \right\}, \tag{3}$$

where the $d_i$ are differentiable with respect to $u_i'$. The functions $d_i(u_i', u_i)$ can be interpreted as distances that penalize deviations of the counterfactual latent variables to their factual realizations.

Backtracking counterfactuals fulfill an intuitive notion of causal compliance, which we will elaborate on the example shown in Fig. 2. To simplify the exposition, we only consider the modes (see § 3.1.2) of the backtracking variables (denoted by $u_1^*$, $u_2^*$, $u_3^*$, $x_1^*$, $x_2^*$ and $x_3^*$) and note that only the latent variables $U_1^*$ and $U_2^*$, which are upstream (in the causal graph) of the antecedent variable $X_2^*$, can contribute in realizing $x_2^* \neq x_2$. Thus, we generally have $u_1^* \neq u_1$ and $u_2^* \neq u_2$, i.e., the counterfactual modes differ from the factual realizations. The downstream latent variable $U_3^*$, in contrast, does not have a causal influence on $X_2^*$. In order to minimize the distance $d_3(u_3^*, u_3)$, $u_3^*$ is thus left unchanged from the factual $u_3$, i.e., $u_3^* = u_3$. The inequality $x_3^* \neq x_3$ is then solely a consequence of the downstream effect of the antecedent $x_2^* \neq x_2$. We will demonstrate these properties experimentally in Section § 4.

This section concludes by introducing so-called *counterfactual explanations*. This allows us to compare our method against this formulation in Section § 3.2.

## 2.4 Counterfactual Explanations

A wealth of prior work in machine learning is concerned with explaining the prediction $\hat{y}$ of a classifier $f_{\hat{Y}}$ with $\hat{y} \leftarrow f_{\hat{Y}}(x)$ through the generation of a new example $x^*$ which is *close* to $x$, yet predicted as $y^*$, where $y^*$ is a label that differs from the (factual) prediction $\hat{y}$. The intuitive idea is that contrasting $x^*$ with $x$ yields an interpretable answer as to why $x$ is classified as $\hat{y}$ rather than $y^*$. Formally (see Wachter et al. (2017)), $x^*$ can be obtained as the solution of

$$\arg\min_{x'} d_o(x', x) \quad \text{subject to} \quad f_{\hat{Y}}(x') = y^*, \tag{4}$$

where $d_o$ represents a distance function between observed variables. In the present work, we generally refer to methods implementing a variant of (4) as *counterfactual explanations*. We stress that $\hat{Y}$ is the prediction of a model ($f_{\hat{Y}}$) and thus always an effect of $X$. In general, the structural assignment $f_{\hat{Y}}$ is different from $f_Y$ (the assignment of the *true* variable $Y$ that is not predicted). For instance, $Y$ might be a *cause* of $X$ or might be *confounded* with $X$. We revisit the difference between $Y$ and $\hat{Y}$ in Section § 3.2.

# 3 Deep Backtracking Counterfactuals (DeepBC)

The main contribution of the present work is to derive formulations and algorithms to efficiently compute backtracking counterfactuals for deep SCMs. In Section § 3.1, we lay down the objectives underlying the two variants of DeepBC that we propose in the present work: (i) *stochastic DeepBC* (§ 3.1.1) aims at sampling from a counterfactual distribution; (ii) *mode DeepBC* (§ 3.1.2) constrains the counterfactual distribution to its mode, thus (deterministically) generating only a single solution. We provide rigorous derivations of the formulations in Section § 3.1 from the theoretical formalization given by von Kügelgen et al. (2023b) in App. A.2. We propose practical algorithms for attaining solutions to the given objectives in Section § 3.3.

## 3.1 Objectives

### 3.1.1 Objective for Stochastic DeepBC

We sample from a distribution over counterfactuals $\mathbf{X}^* \mid \mathbf{x}_S^*, \mathbf{x}$ for the factual realization $\mathbf{x}$, antecedent $\mathbf{x}_S^*$ and (known) backtracking conditional $p^B$.[3] This distribution is characterized by the density

$$p(\mathbf{x}' \mid \mathbf{x}_S^*, \mathbf{x}) \ \propto \ \delta_{\mathbf{x}_S^*}(\mathbf{x}_S') \prod_{i=1}^{n} p_i^B(\mathbf{F}_i^{-1}(\mathbf{x}') \mid \mathbf{F}_i^{-1}(\mathbf{x})) \ \propto \ \delta_{\mathbf{x}_S^*}(\mathbf{x}_S') \exp\left\{ -\sum_{i=1}^{n} d_i(\mathbf{F}_i^{-1}(\mathbf{x}'), \, \mathbf{F}_i^{-1}(\mathbf{x})) \right\}, \quad (5)$$

where $\delta_{\mathbf{x}_S^*}(\cdot)$ refers to the dirac delta at $\mathbf{x}_S^*$. Intuitively, we can understand this distribution as describing counterfactuals that are *likely given* $\mathbf{x}$ in terms of latent components $(p^B)$, while fullfilling the constraint of being compliant both with the antecedent $\mathbf{x}_S^*$ $(\delta_{\mathbf{x}_S^*})$ and with the causal laws $\mathbf{F}$. We further note that (5) is equivalent to a sampling problem within the structured latent space, i.e.,

$$p(\mathbf{u}' \mid \mathbf{x}_S^*, \mathbf{u}) \ \propto \ \delta_{\mathbf{x}_S^*}(\mathbf{F}_S(\mathbf{u}')) \prod_{i=1}^{n} p_i^B(u_i' \mid u_i) \ \propto \ \delta_{\mathbf{x}_S^*}(\mathbf{F}_S(\mathbf{u}')) \exp\left\{ -\sum_{i=1}^{n} d_i(u_i', \, u_i) \right\}. \quad (6)$$

The distribution of $\mathbf{X}^* \mid \mathbf{x}_S^*, \mathbf{x}$ in (5) is obtained as the push-forward of (6) by $\mathbf{F}$, for $\mathbf{x} = \mathbf{F}(\mathbf{u})$.

### 3.1.2 Objective for Mode DeepBC

We compute the mode of $p(\mathbf{x}' \mid \mathbf{x}_S^*, \mathbf{x})$ in (5), i.e., a single "most likely" counterfactual $\mathbf{x}^*$ for the factual realization $\mathbf{x}$ as a solution to the following constrained optimization problem:

$$\arg\min_{\mathbf{x}'} \ \sum_{i=1}^{n} d_i\left(\mathbf{F}_i^{-1}(\mathbf{x}'), \ \mathbf{F}_i^{-1}(\mathbf{x})\right) \qquad \text{subject to} \qquad \mathbf{x}_S' \ = \ \mathbf{x}_S^*. \quad (7)$$

Intuitively, we can understand this optimization as finding a solution $\mathbf{x}^*$ that is *close* to the factual realization $\mathbf{x}$ in terms of its latent components. This situation is visualized on the Morpho-MNIST example in Fig. 1. We further note that (7) is equivalent to an optimization problem within the structured latent space,

$$\arg\min_{\mathbf{u}'} \ \sum_{i=1}^{n} d_i\left(u_i', \ u_i\right) \quad \text{subject to} \quad \mathbf{F}_S(\mathbf{u}') \ = \ \mathbf{x}_S^*, \quad \mathbf{u} = \mathbf{F}^{-1}(\mathbf{x}). \quad (8)$$

We obtain the solution of (7) by inserting the solution of (8) into $\mathbf{F}$.

## 3.2 Relation to Counterfactual Explanations

We can recover counterfactual explanations (§ 2.4) as a special form of mode DeepBC (7). To this end, we assume access to two variables with the following structural equations

$$X \leftarrow f_X(U_X) \quad \text{and} \quad \hat{Y} \leftarrow f_{\hat{Y}}(X), \quad (9)$$

---

[3]In contrast to the reduced form, the backtracking conditional is not learned from data, but must be specified explicitly.

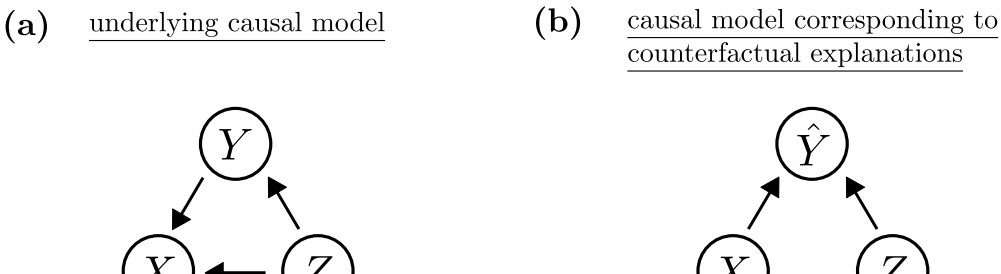

Figure 3: **Backtracking interpretation of counterfactual explanations on a concrete example.**
DeepBC aims at modeling the true structural relationships between variables, exemplified by the causal graph
in **(a)**. Counterfactual explanations in the sense of Wachter et al. (2017) have a backtracking interpretation
in that they instead use a predictive model $f_{\hat{Y}}$ such as a classifier or regressor as structural equation (9),
leading to the causal graph shown in **(b)**. In general, the *true* variable $Y$, unlike the prediction $\hat{Y}$, may
not be the effect of the covariates $X$ and $Z$ ($X$ and $Z$ may in addition be causally interrelated, as shown
in **(a)**). Consequently, the counterfactuals made by counterfactual explanation methods must be interpreted
differently in comparison to those made by our approach. Specifically, DeepBC intents to *explain* the true
underlying variables rather than being confined to the prediction of a model, as can be read off of the
counterfactual queries in the figure. For clarity, the latent variables $\mathbf{U}$ are omitted.

where we note that $\hat{Y}$ is not subject to additional randomness $U_{\hat{Y}}$. In this specific case, we observe that the
mode DeepBC optimization problem (7) reduces to

$$\arg \min_{x'} d_X \left( f_X^{-1}(x'), \ f_X^{-1}(x) \right) \quad \text{subject to} \quad f_{\hat{Y}}(x') = y^*, \tag{10}$$

which can be interpreted as an instance of (4), where distance is measured in an unstructured latent space,
governed by $f_X$. Under the assumption that $f_X$ is modeled as a deep invertible generative model (§ 2.2),
we the refer to (10) as *deep counterfactual explanation*. For example, $x$ could be a high-dimensional image,
$f_X$ an (unconditional) variational autoencoder and $y$ a label of the image.[4] From this viewpoint, we can
interpret DeepBC as a general form of counterfactual explanations in two ways: Firstly, it accommodates
non-deterministic relations among variables, taking into account the influence of noise on all variables. In
the aforementioned instance (9), this can be modeled by $\hat{Y} \leftarrow f_{\hat{Y}}(X, U_{\hat{Y}})$ (demonstrated in Fig. 12 **(b)**
of App. D.2). Secondly, DeepBC accounts for multiple variables with more general underlying causal
relationships. For example, there could be a third variable $Z$ related to both $X$ and $Y$ in (10) that could
be modeled as well, as depicted in Fig. 3 **(a)**. Rather than treating $Z$ as a new dimension in the predictor
input $f_{\hat{Y}}$ in (9) (resulting in $f_{\hat{Y}}(X, Z)$, see Fig. 3 **(b)**), DeepBC allows for explicitly modeling the true
causal relations between $Z$, $X$ and $Y$ (see Fig. 3 **(a)**) via (deep) structural equations, as outlined in Section
§ 2.1. The benefit of DeepBC is that it intends to answer queries regarding the true underlying variables
(see query in Fig. 3 **(a)**) rather than being confined to model predictions that generally do not model the
causal mechanisms according to the true relationships (see query in Fig. 3 **(b)**). We elaborate further on
the benefits of our approach in Section § 3.5.

In the next section (§ 3.3), we propose algorithms to generate backtracking counterfactuals in practice, based
on the formulations presented in (5) and (7).

---

[4]Typically, such methods do not explicitly model other variables besides $x$ and $y$.

| **Algorithm 1** `mode_DeepBC` | **Algorithm 2** `stochastic_DeepBC` |
|---|---|
| **Require:** $\mathbf{x}$, $\mathbf{x}_S^*$, $\mathbf{W}$, $\mathbf{F}$, $\lambda$, $T$ | **Require:** $\mathbf{x}$, $\mathbf{x}_S^*$, $\mathbf{W}$, $\mathbf{F}$, $\lambda$, $T$, $\eta$ |
| $\quad \mathbf{u}'[0] \leftarrow \mathbf{F}^{-1}(\mathbf{x})$ | $\quad \mathbf{u}'[0] \leftarrow \texttt{mode\_DeepBC}(\mathbf{x}, \mathbf{x}_S^*, \mathbf{W}, \mathbf{F}, \lambda, T)$ |
| $\quad$ **for** $t = 0, 1, ..., T-1$ **do** | $\quad$ **for** $t = 0, 1, ..., T-1$ **do** |
| $\quad\quad \bar{\mathbf{J}}_S \leftarrow \mathbf{J}_S(\mathbf{u}'[t])$ | |
| $\quad\quad \tilde{\mathbf{x}}_S^* \leftarrow \mathbf{x}_S^* + \bar{\mathbf{J}}_S \mathbf{u}'[t] - \mathbf{F}_S(\mathbf{u}'[t])$ | $\quad\quad \mathbf{b}[t] \sim \mathcal{N}(\mathbf{0}, \mathbf{I})$ |
| $\quad\quad \mathbf{u}'[t+1] \leftarrow (\mathbf{W} + \lambda \bar{\mathbf{J}}_S^\top \bar{\mathbf{J}}_S)^{-1}(\mathbf{W}\mathbf{u} + \lambda \bar{\mathbf{J}}_S^\top \tilde{\mathbf{x}}_S^*)$ | $\quad\quad \mathbf{u}'[t+1] \leftarrow \mathbf{u}'[t] - \eta \nabla_{\mathbf{u}'} \mathcal{L}(\mathbf{u}'[t]; \mathbf{u}, \mathbf{x}_S^*) + \sqrt{2\eta}\, \mathbf{b}[t]$ |
| $\quad$ **end for** | $\quad$ **end for** |
| $\quad$ **return** $\mathbf{u}'[T]$ | $\quad$ **return** $\mathbf{u}'[T]$ |

### 3.3 Algorithms

We rely on a penalty formulation to approximate (6) and (8) in order to account for the dirac measure and the constraint, respectively. Specifically, we consider the following energy (loss) function with respect to $\mathbf{u}'$:

$$\mathcal{L}(\mathbf{u}'; \mathbf{u}, \mathbf{x}_S^*) \;:=\; \sum_{i=1}^{n} d_i(u_i', \, u_i) \;+\; \lambda \left\| \mathbf{F}_S(\mathbf{u}') - \mathbf{x}_S^* \right\|_2^2, \tag{11}$$

where $\lambda > 0$ is a sufficiently large penalty parameter and $\mathbf{u} = \mathbf{F}^{-1}(\mathbf{x})$.

#### 3.3.1 Algorithm for Stochastic DeepBC

We propose to sample counterfactuals from $\mathbf{X}^* \,|\, \mathbf{x}_S^*, \mathbf{x}$ by leveraging Langevin Monte Carlo (Parisi, 1981), and therefore consider the time-dependent variable $\mathbf{U}'(t)$, generated by the stochastic differential equation

$$d\mathbf{U}'(t) = -\nabla_{\mathbf{u}'} \mathcal{L}(\mathbf{U}'(t); \mathbf{u}, \mathbf{x}_S^*) dt + \sqrt{2}\, d\mathbf{B}(t), \tag{12}$$

where $\mathbf{B}(t)$ denotes Brownian motion. It can be shown that $\mathbf{U}'$ admits a stationary distribution with probability density

$$p_\infty(\mathbf{u}') \propto \exp\left\{ -\mathcal{L}(\mathbf{u}'; \mathbf{u}, \mathbf{x}_S^*) \right\},$$

and hence we can use (12) to generate approximate samples from the desired distribution (6). In practice, we apply an Euler-Maruyama discretization (Sauer, 2013) of (12). It is given as

$$\mathbf{U}'[t+1] \leftarrow \mathbf{U}'[t] - \eta \nabla_{\mathbf{u}'} \mathcal{L}(\mathbf{U}'[t]; \mathbf{u}, \mathbf{x}_S^*) + \sqrt{2\eta}\, \mathbf{B}[t], \qquad \mathbf{B}[t] \overset{i.i.d.}{\sim} \mathcal{N}(\mathbf{0}, \mathbf{I}), \tag{13}$$

for some step size $\eta > 0$ and discrete time steps denoted by square brackets. The stationary distribution of $\mathbf{U}'$ does not depend on the initialization, which is why we initialize the sampling algorithm (13) at the mode of $\mathbf{U}^* \,|\, \mathbf{F}^{-1}(\mathbf{x}), \mathbf{x}_S^*$, generated by mode DeepBC (see § 3.3.2). The algorithm for generating a single sample is specified in Alg. 2.

#### 3.3.2 Algorithm for Mode DeepBC

We approximately compute the mode of $\mathbf{X}^* \,|\, \mathbf{x}_S^*, \mathbf{x}$ by directly minimizing the energy $\mathcal{L}(\mathbf{u}'; \mathbf{u}, \mathbf{x}_S^*)$ (11) with respect to $\mathbf{u}'$, where $\mathbf{u} = \mathbf{F}^{-1}(\mathbf{x})$. Rather than performing gradient descent on the original objective, we empirically observe that using a first-order Taylor approximation of $\mathbf{F}_S$ at $\bar{\mathbf{u}}$ is beneficial when minimizing the distance

$$d_i(u_i', u_i) = w_i \left\| u_i' - u_i \right\|_2^2$$

with $w_i > 0$. Specifically, we consider the approximation

$$\mathbf{F}_S(\mathbf{u}') \approx \mathbf{F}_S(\bar{\mathbf{u}}) + \mathbf{J}_S(\bar{\mathbf{u}})(\mathbf{u}' - \bar{\mathbf{u}}),$$

where $\bar{\mathbf{u}}$ is the linearization point and $\mathbf{J}_S(\bar{\mathbf{u}}) := \nabla_{\mathbf{u}} \mathbf{F}_S(\bar{\mathbf{u}})^\top$ denotes the Jacobian matrix. As a result of this approximation, (11) is a convex quadratic function in $\mathbf{u}'$ and can therefore be solved for its minimum $\hat{\mathbf{u}}^*$ in closed form:

$$\hat{\mathbf{u}}^* \;=\; (\mathbf{W} + \lambda \mathbf{J}_S^\top(\bar{\mathbf{u}})\mathbf{J}_S(\bar{\mathbf{u}}))^{-1}(\mathbf{W}\mathbf{u} + \lambda \mathbf{J}_S^\top(\bar{\mathbf{u}})\tilde{\mathbf{x}}_S^*), \tag{14}$$

where

$$\tilde{\mathbf{x}}_S^* := \mathbf{x}_S^* + \mathbf{J}_S(\bar{\mathbf{u}})\bar{\mathbf{u}} - \mathbf{F}_S(\bar{\mathbf{u}})$$

and $\mathbf{W} := \operatorname{diag}(w_i)$ is a diagonal matrix containing the distance weights $w_i$. By default, we set $w_i = 1$, for all $i$. A different choice of weights may be useful to encode the notion that certain noise variables are more stable (where large $w_i$ corresponds to a high stability) due to application-specific requirements. A detailed derivation of (14) is provided in App. A.3.

Solving (14) once, starting from the initial condition $\bar{\mathbf{u}} = \mathbf{u}$, does not accurately fulfill the constraint due to the constraint linearization, except for special cases. We thus apply an iterative algorithm similar to the Levenberg-Marquardt method (e.g., Moré (2006)), based on (14) that is specified in Alg. 1. Empirically, we observe Alg. 1 to converge after much fewer iterations than gradient descent algorithms (see App. B.1 for more implementation details and experiments).

### 3.4 Extensions

**Categorical Variables.** The main challenge presented by categorical variables is that parameterizations which are both invertible and differentiable in $U_i$ are not straightforward to obtain. To address this circumstance, we propose an approach that is roughly inspired by the reparameterization trick for discrete variables (Jang et al., 2017). For $K$ classes, we assume that $x_i$ corresponds to a one-hot vector with $x_i^{(k)} = 1$ for its realized class $k$ and $x_i^{(l)} = 0$, for all $l \neq k$. We then approximate the distribution of $X_i$ as follows:

$$X_i^{(k)} \,|\, \mathbf{x}_{\mathrm{pa}(i)} \approx f_i^{(k)}(\mathbf{x}_{\mathrm{pa}(i)}, U_i) := \begin{cases} \dfrac{\exp\left\{g^{(k)}(\mathbf{x}_{\mathrm{pa}(i)}, U_i)/\tau\right\}}{\exp\{c/\tau\} \,+\, \sum_{l=1}^{K-1}\exp\left\{g^{(l)}(\mathbf{x}_{\mathrm{pa}(i)}, U_i)/\tau\right\}}, & \text{if } k \in \{1, ..., K-1\}, \\[4mm] \dfrac{\exp\{c/\tau\}}{\exp\{c/\tau\} \,+\, \sum_{l=1}^{K-1}\exp\left\{g^{(l)}(\mathbf{x}_{\mathrm{pa}(i)}, U_i)/\tau\right\}}, & \text{if } k = K. \end{cases} \tag{15}$$

where $c > 0$ is a constant and $\tau > 0$ is a temperature parameter. The smaller we choose $\tau$, the better the approximation in (15) becomes. The function $g$ corresponds to a conditional normalizing flow that was trained on the logits output by a classifier, obtained either by regressing on $\mathbf{x}_{\mathrm{pa}(i)}$ or $x_{\mathrm{img}}$ (such as the MNIST image in Fig. 1). We see that $f_i$ indeed fulfills the conditions of being both invertible and differentiable in $U_i$, thus enabling the application of DeepBC.

**Sparsity.** We further employ a variant of DeepBC that encourages sparse solutions, where sparsity is measured in $\mathbf{u}$ rather than $\mathbf{x}$. Specifically, we use sparse DeepBC to obtain solutions where only few elements in $\mathbf{u}^*$ differ from $\mathbf{u}$, i.e.,

$$d_i(u_i', u_i) = \|u_i' - u_i\|_0 \,,$$

for all $i$, where $\|\cdot\|_0$ denotes the number of nonzero elements. We apply a greedy approach similar to the one presented in Mothilal et al. (2020), where we start by fixing an integer $M > 0$ for which we desire that $\|\mathbf{u}' - \mathbf{u}\|_0 \leq M$. We then apply DeepBC twice: In a first step, we solve for $\mathbf{u}^*$ using mode DeepBC. Then, we use the $M$ elements of the solution vector with largest $\|u_i' - u_i\|_2$ and apply DeepBC again only on these elements, while fixing the others to $u_i$.

**Other distance functions.** In general, any combination of differentiable distance functions $d_i$ (3) can be used when applying gradient descent-based methods for mode DeepBC.

### 3.5 Properties in the Context of Counterfactual Explanations

We highlight the main contributions of our work in the context of counterfactual explanations, which we demonstrate experimentally in Section § 4 and App. D:

1. **Causal Compliance.** By construction, DeepBC generates counterfactuals that adhere to causal laws $(f_1, f_2, ..., f_n)$ by ensuring the preservation of these laws during the generation process: It delineates similarity between data points $\mathbf{x}, \mathbf{x}'$ in terms of their latent representations $\mathbf{u}, \mathbf{u}'$ that correspond to the pull-back through the causal laws as encoded by the reduced form, i.e., $\mathbf{u} = \mathbf{F}^{-1}(\mathbf{x})$ and $\mathbf{u}' = \mathbf{F}^{-1}(\mathbf{x}')$. This way, the laws are guaranteed to be retained in the generated counterfactuals, independent of the used backtracking conditional. This contrasts counterfactual explanations methods (4), which do not model the true causal mechanisms explicitly and thus may violate causal relationships (see Fig. 4 **(b)** and Fig. 6).

2. **Versatility.** DeepBC naturally supports dealing with complex causal relationships between multiple variables that are potentially high dimensional (e.g., images or scalar attributes). This goes beyond the instance-label setup (4) presented in § 2.4, and thus naturally supports flexible choices of antecedent variables (Fig. 12 **(a)**). This contrasts counterfactual explanation methods that are typically limited to antecedents with respect to one single output of a predictive model. DeepBC further allows for sampling (§ 3.1.1) and varying the distance functions $d_i$ in (7) to obtain counterfactuals with different properties, such as noise preservation (Fig. 11 in App. D.1) or sparsity (Fig. 6).

3. **Modularity.** The structural equations or mechanisms $(f_1, f_2, ..., f_n)$ encoding the causal relations between variables may change across different domains, for example, as the result of interventions or environmental changes. It has been postulated that such changes tend to manifest *sparsely*, meaning that only a few of the modules $f_i$ change at a time, while the others remain fixed (Schölkopf et al., 2021; Perry et al., 2022). By explicitly modeling the individual structural equations (as deep generative components), DeepBC exhibits a natural modularity. As a result, adapting to new domains only requires adjusting those components which undergo a domain shift while all other modules can be re-used. This contrasts with counterfactual explanation methods (4), which do not incorporate such replaceable modules and thus require relearning the entire model to handle domain shifts.

## 4 Experiments

In our experiments, we contrast DeepBC with existing ideas and showcase the properties and abilities outlined in Section § 3.5. As showing a single example per counterfactual is more illustrative than sampling multiple examples, we mainly focus on mode DeepBC (§ 3.1.2, § 3.3.2) and refer to this variant when using the term *DeepBC* from this point on. When referring to stochastic DeepBC, we always explicitly write *stochastic DeepBC*. Technical details about the implementation are provided in App. B.

### 4.1 Morpho-MNIST

**Experimental Setup.** We use Morpho-MNIST, a modified version of MNIST proposed by Castro et al. (2019), to showcase how deep backtracking contrasts with its interventional counterpart (Pawlowski et al., 2020) and how the generated results depend on the correct causal graph (§ 2.1). The data set consists of three variables, two continuous scalars and an MNIST image of a handwritten digit, which all correspond to the observable variables (see § 2.1), depicted in Fig. 1. The first scalar variable $T$ describes the thickness and the second variable $I$ describes the intensity of the digit. They have a non-linear relationship and are positively correlated, as can be seen in Fig. 4 **(a) (ii)**, where the observational density of thickness and intensity is shown in blue. The known causal relationship between thickness and intensity is depicted in Fig. 4 (top left); the true structural equations are listed in App. C. We train a normalizing flow for thickness and one for intensity conditionally on thickness, and model the image given $T$ and $I$ via a conditional $\beta$-VAE (Higgins et al., 2017). We here use $d_i(u_i', u_i) = w_i \|u_i' - u_i\|_2^2$ with $w_i = 1$, $\forall i$ as the default distance function and note that the $u_i$ correspond to $u_T$, $u_I$ and $u_{\text{img}}$.

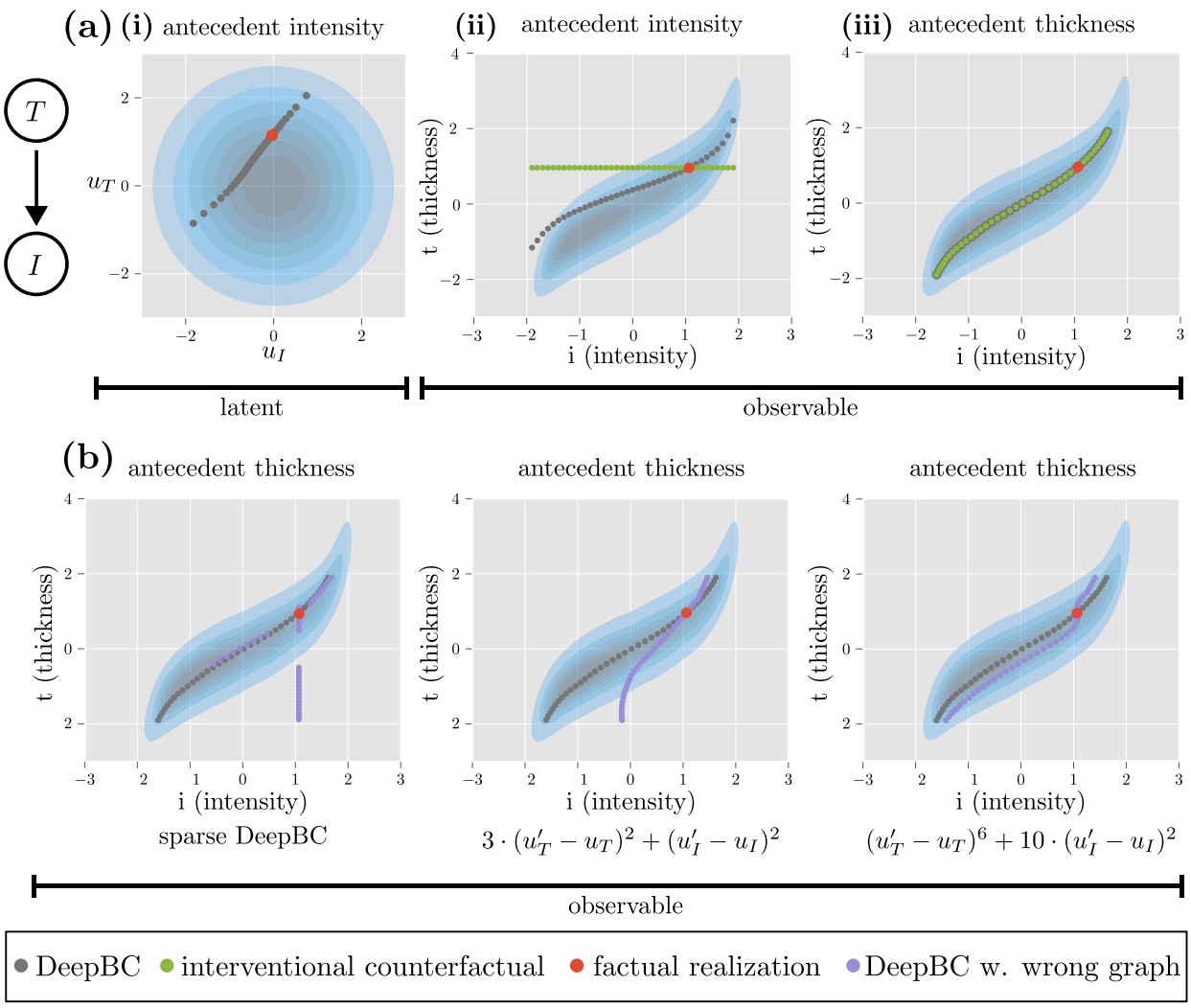

Figure 4: **Counterfactual Scalar Variables on MorphoMNIST.** The blue shaded areas indicate the probability density of the data. **(a) (i)** Given a factual realization (red dot), varying the values of the antecedent $i^*$ changes both $u_I^*$ and the upstream variable $u_T^*$. Since interventional counterfactuals do not perturb the latents, only the backtracking solution (grey dots) is shown. **(ii)** Interventional counterfactuals (green dots), in contrast to backtracking counterfactuals, leave $t^*$ unchanged when the effect variable intensity is taken as antecendent. **(iii)** When treating thickness as the antecedent, counterfactual and backtracking counterfactuals yield identical solutions. **(b)** For the correct graph, DeepBC counterfactuals for antecedent thickness do not change as the backtracking conditional (corresponding distance function shown under each subplot) is changed. When we performing DeepBC with the wrong graph $(I \rightarrow T)$, causal compliance as described in § 2.3 is violated.

**Results.** Our results in Fig. 4 **(a)** and Fig. 5 illustrate distinctive properties of the backtracking approach, in comparison to interventional counterfactuals. When choosing the effect variable intensity as the antecedent, backtracking preserves the causal laws and thus changes the upstream (cause) variable thickness accordingly to match the change in intensity as shown in Fig. 4 **(ii)** and Fig. 5. This leads to counterfactuals that resemble images from the original data set, where thickness and intensity change simultaneously, as shown in the top row of Fig. 5. DeepBC arrives at these counterfactuals, since $i^* \neq i$ can either be achieved by choosing a different $u_I^* \neq u_I$ or by changing the upstream latent $u_T^* \neq u_T$ (because $i^*$ also depends on the realization $t^*$, which, in turn, depends on $u_T^*$). As to minimize the sum of squared latent perturbations

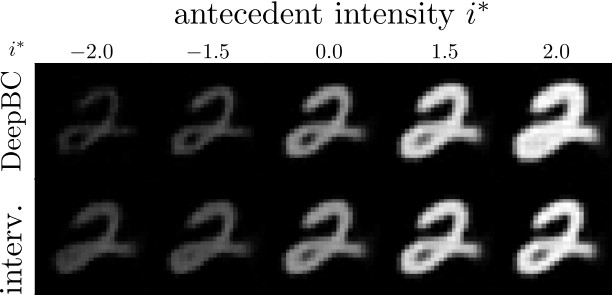

Figure 5: **Counterfactual Morpho-MNIST Images: Backtracking vs. Interventional**. DeepBC (top row) changes intensity alongside thickness, since their causal relation is preserved. Interventional counterfactuals (bottom row), on the contrary, solely change the intensity value, resulting in images that violate the causal laws and can be considered out-of-distribution w.r.t. the original data set.

$d_T(u_T, u_T^*) + d_I(u_I, u_I^*)$, DeepBC tends to change both latent variables from their factual realizations, as can be seen in Fig. 4 **(i)**.

In contrast, the interventional approach breaks the causal link from thickness to intensity when intensity is the antecedent and thus always leaves thickness unchanged, see the green dots in Fig. 4 **(ii)**. This results in counterfactual images with high-intensity but atypically low thickness, or low intensity but typically high thickness, as shown in the second row of Fig. 5. In terms of generating counterfactuals that yield faithful insights into the causal relationships underlying the data, this can be considered a weakness of the interventional approach.

However, interventional and backtracking counterfactuals can also be identical, as shown in Fig. 4 (**iii**), where the thickness variable $T$ is used as antecedent. If the antecedent is a root node of the causal graph $G$, as is the case for $T$, the change in $t^* \neq t$ cannot be traced back to any latent variable other than $u_T$, which is why both $u_I^* = u_I$ and $u_{\text{Img}}^* = u_{\text{Img}}$, analogously to interventional counterfactuals. The change in the value $i^*$ as a function of $t^*$ then solely corresponds to the causal effect of $t^*$, for both counterfactuals (Fig. 4 **(iii)**).

DeepBC depends on the reduced form that is learned from data (1), which in turn depends on the causal graph that is assumed (§ 2.1). In Fig. 4, counterfactuals for the true graph are compared to those from a model that was trained in the same way, with the only difference that the arrow from thickness was reversed. For the correct causal graph ($T \rightarrow I$), we observe that the counterfactuals for antecedent thickness must be invariant with respect to the choice of backtracking conditional (corresponding distance functions are shown below each subplot). This is because intensity is downstream of thickness and so (as mentioned in the previous paragraph and described in § 2.3) it must hold that $u_I^* = u_I$, for any choice of backtracking conditional. However, when using the wrong causal graph ($I \rightarrow T$), we see that the solutions are different and causally incompliant (in the sense of § 2.3). This is because $U_I$ is causally upstream of $T$ in the wrong graph and thus $u_I^*$ contributes to the realization of the antecedent $t^*$, in ways that depend on the choice of the backtracking conditional.

As presented in Section § 3, DeepBC further supports sampling via stochastic DeepBC (§ 3.1.1, § 3.3.1), which is demonstrated in Fig. 10. We also present additional experimental results for antecedent intensity using mode DeepBC in Fig. 11, using different distance functions and weightings (§ 3.4).

## 4.2   CelebA

**Experimental Setup.**   We also investigate generating counterfactual celebrity images on the CelebA data set (Liu et al., 2015). The images have a resolution of $128 \times 128$ and are annotated with binary attributes {Age, Gender, Beard, Bald}. Both the image and the attributes correspond to the observable variables. We adopt the causal graph assumed by Yang et al. (2021b), which is shown in Fig. 6 (top right). We focus on generating counterfactuals that manipulate the considered variables sparsely. Since our optimization algorithms assume differentiability of **F** in **u** (§ 3.3), we preprocess the data to use the standardized logits of classifiers that we trained to predict each attribute from the image (§ 3.4). Analogously to § 4.1, we

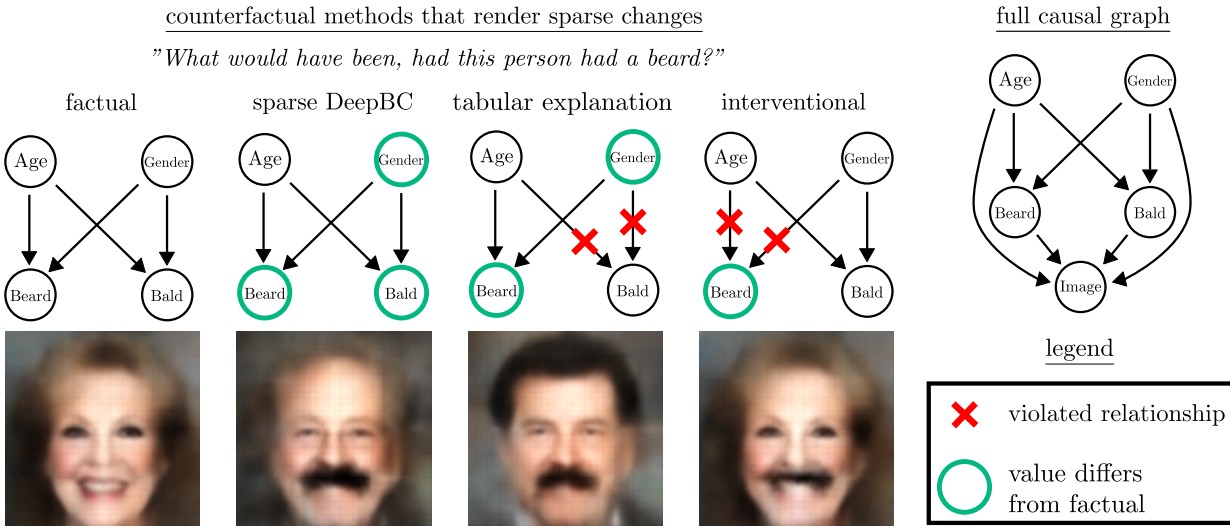

Figure 6: **DeepBC for CelebA**. DeepBC preserves causal relationships, in contrast to other methods: Both sparse DeepBC and the endogenous sparsity method alter gender to add a beard to the elderly woman of the factual realization, while keeping age unchanged. Only sparse DeepBC respects the causal downstream: baldness increases as gender changes. In contrast, the method measuring sparsity in **x** leaves the variable bald unchanged, thereby ignoring the causal relationship by which age and gender affect baldness for a *typical* example.

then train a conditional normalizing flow for each attribute given its parents in the causal graph, and use a conditional $\beta$-VAE for the image given all four attributes.

**Baselines & Ablations.** In addition to comparing DeepBC with the interventional approach, we also consider the following baselines and ablations in generating sparse counterfactuals:

1) **Tabular non-causal explanation:** Prior work in the field of counterfactual explanations have measured distance directly in terms of **x** (4), typically for tabular data such as the low-dimensional attributes of the CelebA data set (e.g., Mothilal et al., 2020; Lang et al., 2023). In the style of these (non-causal) methods (see (4)), we train a new regressor that predicts an attribute from all other attributes (not including the image). We do so for each attribute separately. We then employ sparse DeepBC (§ 3.4) on this regressor, but measure distance directly in the attributes of **x**, instead of measuring distance in **u**. In order to generate a counterfactual image in Fig. 6, we use the conditional variational auto-encoder that was trained for DeepBC and condition the auto-encoder on the counterfactual attribute values, while keeping the realizations $u_{\text{Img}}$ from the factual example, i.e., $u_{\text{Img}}^* = u_{\text{Img}}$.

2) **Wrong causal graph:** We assess how choosing a different causal graph (see Fig. 12 **(d)** in App. D.2) changes the result of the counterfactual. This baseline assesses the impact of model misspecification on the result.

We note that deep counterfactual explanation methods (10) are not applicable for rendering sparse changes, because they cannot identify the underlying attribute variables that the counterfactuals should be sparse in.

**Results.** DeepBC preserves all causal relationships, in contrast to other methods: Fig. 6 shows sparse DeepBC (sparsity threshhold $M = 2$, see § 3.4) and other approaches that are able to generate counterfactuals that render sparse changes with respect to the considered attributes. As can be seen from the causal graph, the elderly woman from the factual image could develop a beard by changing gender and age. Both the sparse tabular explanation method and sparse DeepBC choose only gender (it is much more dependent on beard than on age), leaving the value of age fixed. For sparse DeepBC, despite the latent variable $u_{\text{Bald}}$

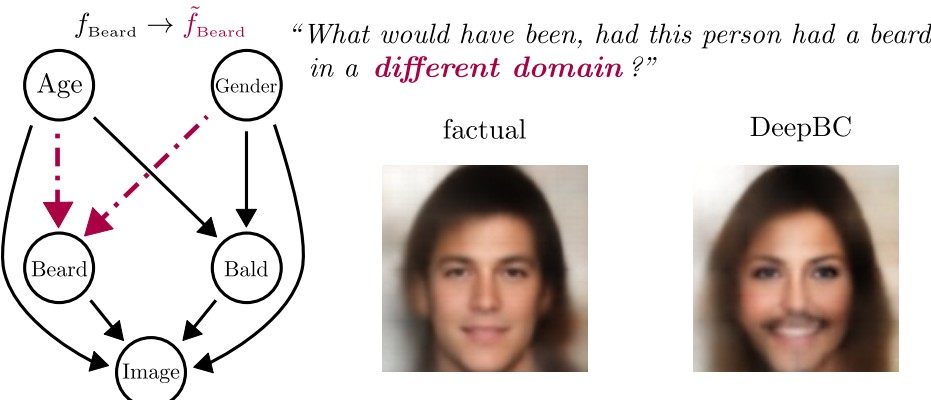

Figure 7: **Modularity of DeepBC.** DeepBC allows for exchanging causal mechanisms, hence allowing for out-of-distribution counterfactuals. In this example, the learned mechanism by which gender and age affect beard ($f_{\text{Beard}}$) is exchanged for a manually constructed one where being female is highly associated with having a beard ($\tilde{f}_{\text{Beard}}$).

*"What would have been, had this person been male?"*

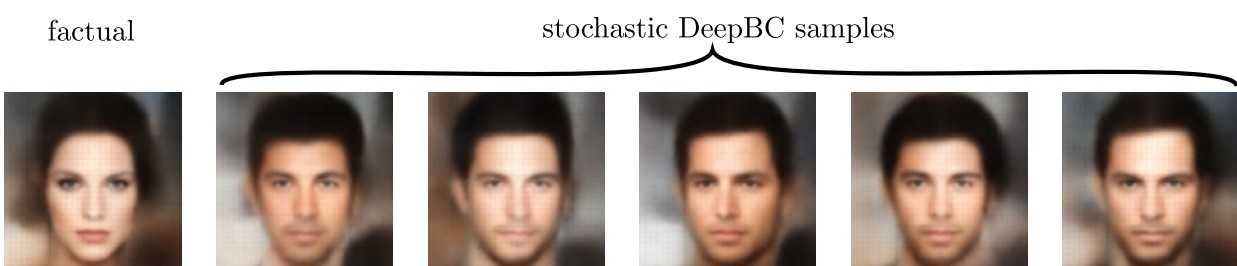

Figure 8: **Stochastic DeepBC for CelebA.** Five counterfactual image samples, generated using stochastic DeepBC (§ 3.1.1, § 3.3.1). It can be seen that the approach facilitates the generation of a diverse set of counterfactuals that fulfill the antecedent condition reliably and resemble the factual image.

not being updated, the realization of bald $x_{\text{Bald}}$ is automatically modified as a downstream effect as encoded by the SCM (being old and male often leads to baldness). This lies in contrast to measuring sparsity in terms of $\mathbf{x}$ directly, without the causal model, where this causal relationship is not taken into account. As a result, the factual value of bald is kept unchanged for the tabular explanation method (baseline 1).

DeepBC is inherently modular, as demonstrated in Fig. 7.[5] In this figure, the mechanism by which age and gender affect beard is manually replaced by a different one. The resulting counterfactual can be interpreted as an *out-of-distribution* counterfactual, because this replaced mechanism does not correspond to the one that was learned from the *in-distribution* data.

In App. D.2.1, we further demonstrate the property of DeepBC to take into account antecedents that consist of more than a single variable. We also demonstrate quantitative experiments and their results in App. D.2.2.

## 5 Related Work

This section is organized into two lines of prior work. The first line encompasses methods that incorporate causality into the field of counterfactual explanations. However, we note that the general field of counterfactual explanations has made many significant advances that are not directly related to causality. For comprehensive overviews over these developments, we refer for example to Guidotti (2022) and Verma et al.

---

[5]This contrasts with counterfactual explanation methods specifically.

(2020). The second line discusses how deep neural networks have been used within the context of SCMs, as to facilitate counterfactual computation.

**Causality in Counterfactual Explanations.** A prominent line of work raises the importance of causality to ensure actionability of counterfactual explanations in a sense that an alternative outcome could have been achieved by *performing alternative actions or interventions*, without violating the remaining causal relationships (Karimi et al., 2020; 2021). These works fundamentally differ from ours in that the considered interventions actively break some of the causal relationships, which lies in stark contrast to the backtracking approach (see Fig. 2). The latter seeks to trace back counterfactuals to changes in latent variables rather than changes in causal relationships. However, this line of research is related to ours in that it argues for respecting the causal structure and mechanisms in generating counterfactuals.

The most similar existing work to ours is that of Mahajan et al. (2019). Similarly to the present work, the authors also employ deep generative models and measure the distance between factual and counterfactual examples in a latent space. The most distinctive difference to our work is that Mahajan et al. (2019) impose causal constraints via a *causal proximity loss* in the observable variables $\mathbf{x}$ that assumes additive Gaussian noise, thereby restricting the types of causal mechanisms $f_i$ under consideration. The *causal proximity loss* approach is at odds with the backtracking philosophy (Lewis, 1979) that we follow. In our approach, all changes are traced back solely to latent variables $\mathbf{u}$ that are embedded into the deep causal model, such that all causal constraints are satisfied automatically by construction (§ 3.5). This obviates the need for an additional loss and straightforwardly allows for general noise dependencies (see App. C). At the same time, our approach is more versatile as any subset of the given variables could be used as antecedent, whereas the method of Mahajan et al. (2019) only supports a specific label variable (see (4)), similar to the vast majority of counterfactual explanation methods.

**Counterfactuals in Deep Structural Causal Models.** The integration of deep generative components such as normalizing flows and variational auto-encoders into SCMs can be traced back at least to the works of Kocaoglu et al. (2018); Goudet et al. (2018); Pawlowski et al. (2020) and others. Subsequently, this approach has been adopted in various works for computing counterfactuals in applications such as natural language processing (Hu & Li, 2021) and bias reduction (Dash et al., 2022). Other recent works have explored the use of graph neural networks (Sanchez-Martin et al., 2022), normalizing flows (Khemakhem et al., 2021; Javaloy et al., 2023) and diffusion probabilistic models (Sanchez & Tsaftaris, 2022) to construct SCMs.

In the present work, we employ variational auto-encoders and normalizing flows to construct deep SCMs (as outlined in § 2.2). Nevertheless, we regard the design choices within our implementation as orthogonal to various choices of architecture. Specifically, we believe that our approach is applicable to any deep SCM architecture that yields a reduced form which is both (approximately) invertible and differentiable. We believe that this is true because no further assumptions are imposed.

## 6 Discussion & Limitations

**Identifiability of the Reduced Form.** In general, neither the structural equations nor the reduced form (see § 2.1) are identifiable from observational data (Hyvärinen et al., 2024; Karimi et al., 2020; Locatello et al., 2019). However, under certain conditions, it has been shown that the structural equations (and therefore the reduced form) can be identified, up to simple transformations (Javaloy et al., 2023; Nasr-Esfahany et al., 2023). If the causal graph and/or underlying variables are not known either (i.e., we only have access to the high-dimensional images), the problem becomes even worse because both the graph and underlying attributes (e.g., the variables beard, gender, etc. in § 4.2) can also not be identified, in general. Recent works have established numerous conditions under which identifiability of variables and/or graph holds up to certain indeterminancies (e.g., Lachapelle et al., 2022; Buchholz et al., 2023; Lippe et al., 2023; Liang et al., 2023; von Kügelgen et al., 2023a). We view solutions and results to identifying the causal graph and/or the reduced form as complementary to our proposed method and note that the learned mechanisms of DeepBC generally only correspond to approximations of the true mechanisms.

**Non-Invertible Generative Models.**  A potential avenue for future research could be to explore how backtracking could be implemented for generative models whose latent variables cannot be inferred deterministically from the factual realization, such as diffusion probabilistic models (Sohl-Dickstein et al., 2015; Ho et al., 2020) and generative adversarial networks (Goodfellow et al., 2014), both of which are not invertible in general (although approximately invertible variants like DDIMs (Song et al., 2021) do exist). One conceivable solution might be to adapt (6)/(8) as to jointly sample/optimize over $\mathbf{u}$ and $\mathbf{u}'$ or to employ variational inference techniques.

**Model Misspecification.**  The dimensionality of the latent variables for variational autoencoders plays a role in the generated counterfactuals.[6]  Typically, one assumes this dimensionality to be lower than the dimensionality of the data, motivated by the assumption that the data lives on a low-dimensional manifold (Reizinger et al., 2022; Bonheme & Grzes, 2022; Pope et al., 2021). Another source of model misspecification that may render DeepBC counterfactuals incorrect stems from the non-invertibility of the true underlying mechanisms. While this invertibility is a fairly common assumption in the literature (e.g., Pawlowski et al. (2020); Javaloy et al. (2023); Nasr-Esfahany et al. (2023); Hoyer et al. (2008)), it does not hold in general[7] and can also not be ruled out from data alone. A remedy for this restriction may be to extend our method to non-invertible generative models, as outlined above.

**Limitations of Non-Causal Counterfactual Methods.**  The explicit access to the reduced form of a causal model (or at least a good approximation thereof) allows for obtaining causally compliant solutions for varying choices of distance function (§ 3.5, § 4). We note, however, that many non-causal counterfactual methods exist (corresponding to different variants of (4)) that do not rely on knowledge of underlying causal variables and their causal relationships (see e.g., Guidotti (2022)). Based on theoretical results in independent component analysis (Hyvärinen & Pajunen, 1999), we suspect that for certain types of backtracking conditional (e.g., rotationally invariant ones) alongside further assumptions on the generative process (e.g., $x_{\text{img}}$ is a conformal map of all latents $\mathbf{u}$) and dimensionality of $\mathbf{u}$, deep counterfactual explanations (10) on $x_{\text{img}}$ may be able to perform causally compliant backtracking implicitly. In the general setting, however, our empirical results (§ 4) show that this is not the case. This can be made concrete on the Morpho-MNIST example (§ 4.1): Neither the underlying variables intensity and thickness, nor their causal relationship can (in general) be implicitly identified (as discussed in the first paragraph of this section), for example by using a deep counterfactual explanation method applied to the image representation (10). In fact, the experiments in Fig. 4 **(b)** clearly show that backtracking counterfactuals depend on the true causal graph between thickness and intensity.

## 7   Conclusion

In this work, we presented DeepBC, a practical framework for computing backtracking counterfactuals for deep SCMs. We compared DeepBC to interventional counterfactuals and the main formulations employed in the field of counterfactual explanations. We found that, compared to prior work in counterfactual explanations, DeepBC is: compliant with respect to the given causal model; versatile in that it supports unrestricted, complex causal relationships; and modular in that it enables generalization to out-of-distribution settings. In fact, DeepBC can be seen as a general method for computing counterfactuals that measures distances between factual and counterfactual in the structured latent space of an underlying deep causal model. We empirically demonstrated the merits of our approach in comparison to prior work, where we highlight the importance of taking causal relationships into account. We hope that our approach will contribute to future developments of deep explanation methods that provide more faithful insights into the data generating process.

---

[6]For normalizing flows, in contrast to variational autoencoders, the latent space must have the same dimensionality as the observed space.

[7]A simple example is the assignment $X \leftarrow U^2$.

## Reproducibility Statement

Our source code is available at https://github.com/rudolfwilliam/DeepBC. Detailed instructions for reproducing all experiments are provided in the `README.md` file at the top level of the repository. All parameters can be found in the `config` folders within the respective subfolders. In addition, we provide a detailed description of the optimization parameters in App. B.1, training procedures in App. B.3.1 and model architectures in App. B.3.2.

## Acknowledgements

The authors are grateful for valuable discussions with Valentyn Boreiko, Partha Ghosh, and Luigi Gresele. The authors acknowledge the support provided by the German Research Foundation. The authors thank the International Max Planck Research School for Intelligent Systems (IMPRS-IS) for supporting Klaus-Rudolf Kladny.

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

# A   Formalisms & Derivations

## A.1   Formal Definition of Interventional and Backtracking Counterfactuals

Both kinds of counterfactuals can be computed in a three-step-procedure.

__Interventional Counterfactuals__

1. **Abduction**: Compute the distribution of $\mathbf{U} \mid \mathbf{x}$, given the factual realization $\mathbf{x}$ of $\mathbf{X}$.

2. **Action**: Obtain an altered collection of structural assignments $(f_1^*, f_2^*, ..., f_n^*)$ by setting $x_i \leftarrow x_i^* = f_i^*$, for all $i \in S$. Leave all other structural assignments unmodified, i.e., $f_j^* = f_j$, for all $j \notin S$.

3. **Prediction**: Compute a distribution over $\mathbf{X}_{\mathrm{I}}^*$ as the pushforward of the distribution of $\mathbf{U} \mid \mathbf{x}$ by $\mathbf{F}^*$.

__Backtracking Counterfactuals__

1. **Cross-World Abduction**: Use the antecedent $\mathbf{x}_S^*$ and the factual realization $\mathbf{x}$ to obtain $p(\mathbf{u}', \mathbf{u} \mid \mathbf{x}_S^*, \mathbf{x})$, using the backtracking conditional $p^B(\mathbf{u}' \mid \mathbf{u})$ and latent prior density $p(\mathbf{u})$:

$$p(\mathbf{u}', \mathbf{u} \mid \mathbf{x}_S^*, \mathbf{x}) \;=\; \frac{p(\mathbf{u}', \mathbf{u}, \mathbf{x}_S^*, \mathbf{x})}{p(\mathbf{x}_S^*, \mathbf{x})} \;=\; \frac{p^B(\mathbf{u}' \mid \mathbf{u})\, p(\mathbf{u})\, \delta_{\mathbf{x}}(\mathbf{F}(\mathbf{u}))\delta_{\mathbf{x}_S^*}(\mathbf{F}_S(\mathbf{u}'))}{\int \int p^B(\bar{\mathbf{u}}' \mid \bar{\mathbf{u}})\, p(\bar{\mathbf{u}})\, \delta_{\mathbf{x}}(\mathbf{F}(\bar{\mathbf{u}}))\delta_{\mathbf{x}_S^*}(\mathbf{F}_S(\bar{\mathbf{u}}'))\, d\bar{\mathbf{u}}\, d\bar{\mathbf{u}}'},$$

where $\delta_{\mathbf{x}}(\,\cdot\,)$ refers to the dirac delta at $\mathbf{x}$ and $p^B(\,\cdot \mid \cdot\,)$ corresponds to the backtracking conditional (§ 2.3).

2. **Marginalization**: Marginalize over $\mathbf{U}$ to obtain the density $p(\mathbf{u}' \mid \mathbf{x}_S^*, \mathbf{x})$ of the counterfactual posterior:

$$p(\mathbf{u}' \mid \mathbf{x}_S^*, \mathbf{x}) \;=\; \int p(\mathbf{u}', \mathbf{u} \mid \mathbf{x}_S^*, \mathbf{x})\, d\mathbf{u}.$$

3. **Prediction**: Compute a distribution over $\mathbf{X}_{\mathrm{B}}^*$ by marginalizing over the counterfactual latents $\mathbf{U}^*$:

$$p(\mathbf{x}' \mid \mathbf{x}_S^*, \mathbf{x}) \;=\; \int p(\mathbf{u}' \mid \mathbf{x}_S^*, \mathbf{x})\delta_{\mathbf{x}'}(\mathbf{F}(\mathbf{u}'))\, d\mathbf{u}'.$$

## A.2   Formal Derivation of DeepBC

We derive (7) and (5) from the three-step-procedure of backtracking counterfactuals (see App. A.1) as follows:

1. **Cross-World Abduction**: By the deterministic relationship between latents and observables, we see that
$$p(\mathbf{u}', \mathbf{u} \mid \mathbf{x}_S^*, \mathbf{x}) \;=\; p(\mathbf{u}' \mid \mathbf{u}, \mathbf{x}_S^*, \mathbf{x})\, p(\mathbf{u} \mid \mathbf{x}_S^*, \mathbf{x}) \;=\; p(\mathbf{u}' \mid \mathbf{u}, \mathbf{x}_S^*)\, p(\mathbf{u} \mid \mathbf{x})$$
$$= p(\mathbf{u}' \mid \mathbf{u}, \mathbf{x}_S^*)\, \delta_{\mathbf{F}^{-1}(\mathbf{x})}(\mathbf{u}).$$

2. **Marginalization**: All the probability is located at $\mathbf{F}^{-1}(\mathbf{x})$, which is why marginalization reduces to

$$p(\mathbf{u}' \mid \mathbf{x}_S^*, \mathbf{x}) \;=\; p(\mathbf{u}' \mid \mathbf{F}^{-1}(\mathbf{x}), \mathbf{x}_S^*) \;\propto\; p(\mathbf{u}', \mathbf{x}_S^* \mid \mathbf{F}^{-1}(\mathbf{x}))$$

$$= p(\mathbf{x}_S^* \mid \mathbf{u}')\, p(\mathbf{u}' \mid \mathbf{F}^{-1}(\mathbf{x})) \;=\; \delta_{\mathbf{x}_S^*}(\mathbf{F}_S(\mathbf{u}'))\prod_{i=1}^{n} p_i^B(u_i' \mid \mathbf{F}_i^{-1}(\mathbf{x})), \quad (16)$$

where $p(\mathbf{u}' \mid \mathbf{x}_S^*, \mathbf{x})$ corresponds to the density of $\mathbf{U}^* \mid \mathbf{F}^{-1}(\mathbf{x}), \mathbf{x}_S^*$.

3. **Prediction**: By the deterministic relationship between latents and observables, we obtain samples from $\mathbf{X}^* \mid \mathbf{x}_S^*, \mathbf{x}$ simply by sampling from $\mathbf{U}^* \mid \mathbf{F}^{-1}(\mathbf{x}), \mathbf{x}_S^*$ and then subsequently mapping these samples through the function $\mathbf{F}(\mathbf{u}^*)$ to obtain the corresponding observables $\mathbf{x}^*$:

$$\mathbf{u}^* \;\sim\; \mathbf{U}^* \mid \mathbf{F}^{-1}(\mathbf{x}), \mathbf{x}_S^*, \quad \mathbf{x}^* \;=\; \mathbf{F}(\mathbf{u}^*). \tag{17}$$

We observe that (17) corresponds to performing stochastic DeepBC (6).

In order to derive mode DeepBC, we restrict ourselves to the mode of the distribution of $\mathbf{U}^* \mid \mathbf{F}^{-1}(\mathbf{x}), \mathbf{x}_S^*$. We recall that we assume that the backtracking conditional density $p(\mathbf{u}' \mid \mathbf{u})$ has the form

$$p^B(\mathbf{u}' \mid \mathbf{u}) \; \propto \; \exp\left\{ -\sum_{i=1}^{n} d_i(u_i', u_i) \right\},$$

where $d_i$ are distance functions. Then, we have

$$p(\mathbf{u}' \mid \mathbf{F}^{-1}(\mathbf{x}), \mathbf{x}_S^*) \; \propto \; \begin{cases} \exp\left\{ -\sum_{i=1}^{n} d_i\left(u_i', \mathbf{F}_i^{-1}(\mathbf{x})\right) \right\}, & \text{if } \mathbf{F}_S(\mathbf{u}') = \mathbf{x}_S^* \\ 0, & \text{otherwise.} \end{cases}$$

By taking the logarithm and ignoring constants, we obtain

$$\log p(\mathbf{u}' \mid \mathbf{F}^{-1}(\mathbf{x}), \mathbf{x}_S^*) \; = \; \begin{cases} -\sum_{i=1}^{n} d_i\left(u_i', \mathbf{F}_i^{-1}(\mathbf{x})\right), & \text{if } \mathbf{F}_S(\mathbf{u}') = \mathbf{x}_S^* \\ -\infty, & \text{otherwise.} \end{cases}$$

We conclude by noting that $\arg\max_{\mathbf{u}'} \log p(\mathbf{u}' \mid \mathbf{F}^{-1}(\mathbf{x}), \mathbf{x}_S^*)$, composed with $\mathbf{F}$, is equivalent to (7).

### A.3 Derivation of (7)

As a result of the linearization of $\mathbf{F}$, $\mathcal{L}(\mathbf{u}'; \mathbf{u}, \mathbf{x}_S^*)$ in (11) simplifies to

$$(\mathbf{u}' - \mathbf{u})^\top \mathbf{W}(\mathbf{u}' - \mathbf{u}) \; + \; \lambda\|\mathbf{J}_S(\mathbf{u}' - \mathbf{u}) + \mathbf{F}_S(\mathbf{u}) - \mathbf{x}_S^*\|_2^2$$
$$= \; (\mathbf{u}' - \mathbf{u})^\top \mathbf{W}(\mathbf{u}' - \mathbf{u}) \; + \; \lambda\|\mathbf{J}_S\mathbf{u}' - \tilde{\mathbf{x}}_S^*\|_2^2 \; =: \; \tilde{\mathcal{L}}(\mathbf{u}'), \quad (18)$$

where $\tilde{\mathbf{x}}_S^* \coloneqq \mathbf{x}_S^* + \mathbf{J}_S(\mathbf{u})\mathbf{u} - \mathbf{F}_S(\mathbf{u})$. We see that $\tilde{\mathcal{L}}(\mathbf{u}')$ is convex and differentiable with respect to $\mathbf{u}'$, which means that $\nabla_{\mathbf{u}'}\tilde{\mathcal{L}}(\mathbf{u}') = \mathbf{0}$ implies optimality of $\mathbf{u}'$. To derive $\mathbf{u}'_{\mathrm{opt}}$, we observe that

$$\nabla_{\mathbf{u}'}\tilde{\mathcal{L}}(\mathbf{u}') \; = \; 2(\mathbf{W}(\mathbf{u}' - \mathbf{u}) + \lambda\mathbf{J}_S^\top\mathbf{J}_S\mathbf{u}' - \lambda\mathbf{J}_S^\top\tilde{\mathbf{x}}_S^*).$$

As a result, $\mathbf{u}'_{\mathrm{opt}}$ is given by

$$\mathbf{u}'_{\mathrm{opt}} \; = \; (\mathbf{W} + \lambda\mathbf{J}_S^\top\mathbf{J}_S)^{-1}(\mathbf{W}\mathbf{u} + \lambda\mathbf{J}_S^\top\tilde{\mathbf{x}}_S^*).$$

## B Implementation

### B.1 Technical Details and Comments for the DeepBC Optimization Algorithm

In practice, we implement (14) as follows

$$\hat{\mathbf{u}}^* = (\lambda^{-1}\mathbf{W} + \mathbf{J}_S^\top\mathbf{J}_S)^\dagger(\lambda^{-1}\mathbf{W}\mathbf{u} + \mathbf{J}_S^\top\tilde{\mathbf{x}}_S^*), \tag{19}$$

where $\dagger$ denotes Moore-Penrose pseudoinverse. We employ (19) rather than (14) for the reason of numerical stability. The main computational bottleneck in Alg. 1 is the computation of the pseudoinverse $(\lambda^{-1}\mathbf{W} + \mathbf{J}_S^\top\mathbf{J}_S)^\dagger$ in (19), which comes at a cost of $\mathcal{O}(\#\text{it}\cdot\dim(\mathbf{u})^3)$, compared to $\mathcal{O}(\#\text{it}\cdot\dim(\mathbf{u}))$ for gradient descent. We note, however, that the dimensionality of the latent space is typically not very large in our experiments. The maximum dimension is 516 for CelebA, due to 4 attributes and 512-dimensional latent space of the VAE. We also stress that $\mathbf{J}_S$ is sparse (many 0 entries) when $S$ covers attribute variables, because the 512-dimensional latent vector is not upstream of any attribute. We do not run experiments where many variables are upstream of the antecedent variable and stress that this may affect the performance of Alg. 1.

In our experiments, we find Alg. 1 to converge after few iterations, as can be seen in Fig. 9. Typically, convergence can be expected to occur within $\approx 5$ iterations, while fulfilling the constraint reliably (see table in Fig. 9). When applying gradient descent methods like Adam instead of our approach, we observe

that the convergence rate is sensitive to the choice of learning rate. The plot for $\lambda = 10^6$ shows that the linearization method can lead to oscillations if $\lambda$ is chosen too large, which likely stems from small eigenvalues of $\lambda^{-1}\mathbf{W} + \mathbf{J}_S^\top \mathbf{J}_S$ (we not that $\mathbf{J}_S^\top \mathbf{J}_S$ is low-rank) that give rise to numerical issues. However, these oscillations can be detected early on. Similar to Levenberg-Marquardt, we could include a small damping variable $\epsilon > 0$ to alleviate this issue. We would then arrive at

$$\hat{\mathbf{u}}^* = (\lambda^{-1}\mathbf{W} + \mathbf{J}_S^\top \mathbf{J}_S + \mathbf{I}\epsilon)^\dagger (\lambda^{-1}\mathbf{W}\mathbf{u} + \mathbf{J}_S^\top \tilde{\mathbf{x}}_S^*).$$

We do not explore this possibility in the present work as we do not encounter these issues in our experiments.

Adam's convergence highly depends on the choice of learning rate. We suspect that this is due to the poorly conditioned Hessian that comes from choosing large $\lambda$. However, large $\lambda$ is required in order to (at least approximately) fulfill the constraint in (7) (see top row in the table of Fig. 9).

In our experiments, we always use DeepBC via constraint linearization (Alg. 1) with $\lambda = 10^3$ and #it $= 30$. $\lambda$ is chosen empirically and our choice yielded convincing results in all experiments. The choice of iteration number is a conservative upper bound for the algorithm.

## B.2   Implementation Details of Stochastic DeepBC

For both figures Fig. 10 and Fig. 8, we employ Alg. 2 with penalty parameter $\lambda = 10^4$ and for $T = 1000$ iterations. For Fig. 10, we choose step size $\eta = 10^{-5}$ and $w_i = 1, \forall i$. For Fig. 8, we choose $\eta = 10^{-4}$ and $w_i = 1.8, \forall i$.

## B.3   Implementation Details of the Deep Structural Causal Models

For all experiments, we use `PyTorch` (Paszke et al., 2019), `PyTorch Lightning` (Falcon, William and The PyTorch Lightning team, 2019) and `normflows` (Stimper et al., 2023).

### B.3.1   Training Procedures

We train all models with the following parameters:

| optimizer | train/val. split ratio | regularization | max. # epochs |
|:---:|:---:|:---:|:---:|
| Adam | 0.8 | early stopping | 1000 |

**Morpho-MNIST.**   We use the same training parameters for both normalizing flow models. Patience refers to the number of epochs without further decrease in validation loss that early stopping regularization waits.

| model | batch size train | batch size val. | learning rate | patience |
|:---:|:---:|:---:|:---:|:---:|
| **Flow** | 64 | full | $10^{-3}$ | 2 |
| **VAE** | 128 | 256 | $10^{-6}$ | 10 |

**CelebA.**   We use the same training parameters for all normalizing flow models.

| model | batch size train | batch size val. | learning rate | patience |
|:---:|:---:|:---:|:---:|:---:|
| **Flow** | 64 | 256 | $10^{-3}$ | 2 |
| **VAE** | 128 | 256 | $10^{-6}$ | 50 |

### B.3.2   Network Architectures

**Notation.**   We denote concatenations of variables by $[\cdot, \cdot, ..., \cdot]$. We denote modules that are repeated $n$ times by a superscript $(n)$. For instance, Linear$^{(2)}(u)$ is shorthand for Linear $\circ$ Linear $(u)$, i.e., two linear layers.

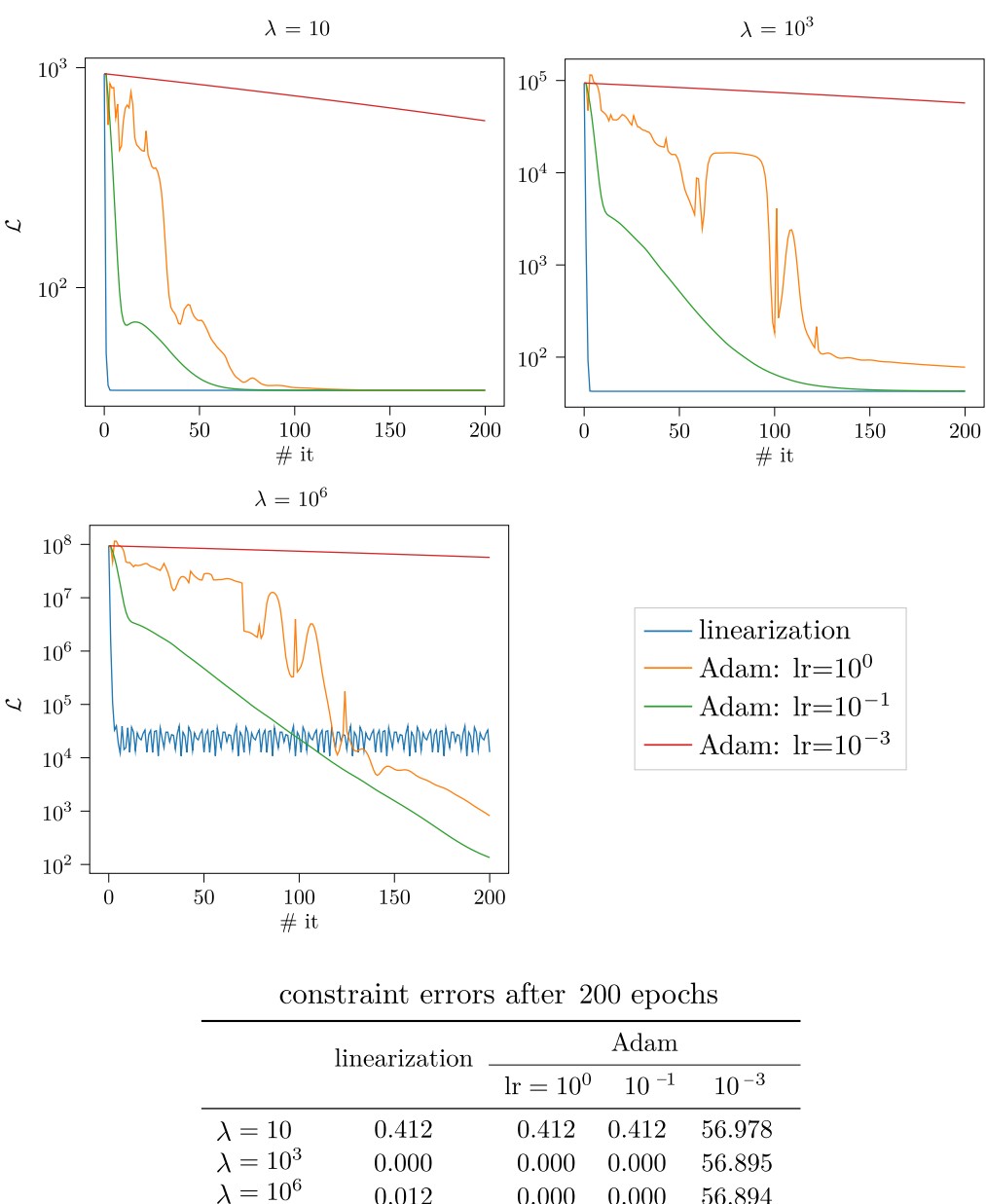

| | linearization | Adam | | |
|---|---|---|---|---|
| | | $lr = 10^0$ | $10^{-1}$ | $10^{-3}$ |
| $\lambda = 10$ | 0.412 | 0.412 | 0.412 | 56.978 |
| $\lambda = 10^3$ | 0.000 | 0.000 | 0.000 | 56.895 |
| $\lambda = 10^6$ | 0.012 | 0.000 | 0.000 | 56.894 |

constraint errors after 200 epochs

Figure 9: The figures show the sum of penalty losses (11) over all points in Fig. 4 **(b)** for optimizing it for 200 iterations on a $\log_{10}$ scale. Comparison of the Adam optimizer with various learning rates in comparison to constraint linearization (Alg. 1) for different choices of penalty parameter $\lambda$. The table shows the sum of constraint errors $\|\mathbf{F}_S(\mathbf{u}') - \mathbf{x}_S^*\|_2^2$ after 200 iterations (How well the constraints are fulfilled).

**Flow Layers.** In all of our experiments, we make use of common types of flow layers:

QuadraticSpline($u_i$) is a standard quadratic spline flow (Durkan et al., 2019).

ConstScaleShift($u_i$) performs a constant affine transformation with learned, but unconditional, location and scale parameters $\mu$ and $\sigma$:

$$\text{ConstScaleShift}(u_i) = \sigma \cdot u_i + \mu.$$

ScaleShift($u_i, \mathbf{x}_{\text{pa}(i)}$) performs the same operation as ConstScaleShift($u_i$), but $\mu$ and $\sigma$ are computed as a function of $\mathbf{x}_{\text{pa}(i)}$ via a two-layer neural network with ReLU activation functions and one-dimensional hidden units.

**Morpho-MNIST.** For the thickness variable, we construct the flow as

$$f_T(u_T) = \text{ConstScaleShift} \circ \text{QuadraticSpline}^{(5)}(u_T).$$

For intensity, we use

$$f_I(t, u_I) = \text{ConstScaleShift} \circ \text{Sigmoid} \circ \text{QuadraticSpline}^{(3)} \circ \text{ScaleShift}([t, u_I]),$$

where Sigmoid denotes the (constant) sigmoid function.

For the MNIST image, we use a convolutional $\beta$-VAE (Higgins et al., 2017) with $\beta = 3$ and the following encoder parameterization:

$$\begin{aligned}
f_{\text{Img}}(t, i, \text{img}) &\approx e_{\text{Img}}(t, i, \text{img}) \\
&= \text{Linear}\left(\left[t,\ i,\ \left(\text{Linear} \circ \text{Pool2D} \circ (\text{ReLU} \circ \text{Conv2D})^{(4)}\right)(\text{img})\right]\right),
\end{aligned}$$

where the Conv2D layers (starting with parameters from the layer closest to the input) are parameterized by `out_channels` $= (8, 16, 32, 64)$, `kernel_size` $= (4, 4, 4, 3)$, `stride` $= (2, 2, 2, 2)$, `padding` $= (1, 1, 1, 0)$. The linear layers are analogously parameterized with the output dimensions `out` $= (128, 16, 16)$, i.e., $\dim(u_{\text{Img}}) = 32$. For the decoder, we use

$$\begin{aligned}
f_{\text{Img}}^{-1}(t, i, u_{\text{Img}}) &\approx d_{\text{Img}}(t, i, u_{\text{Img}}) \\
&= \text{TransConv2D} \circ (\text{ReLU} \circ \text{TransConv2D})^{(4)} \circ \text{Linear}([t, i, u_{\text{Img}}]),
\end{aligned}$$

where the linear layer has output dimension `out` $= 64$ and the transpose convolution layers (starting with parameters from the layer closest to the input) are parameterized by `out_channels` $= (64, 32, 16, 1)$, `kernel_size` $= (3, 4, 4, 4)$, `stride` $= (2, 2, 2, 2)$, `padding` $= (0, 1, 0, 1)$.

**CelebA.** We preprocess all attributes via separate classifiers $C_{\text{Attr}}$, i.e., one individual classifier per attribute. The classifier has the following architecture:

$$C_{\text{Attr}}(\text{img}) = \text{Linear} \circ \text{Dropout} \circ \text{ReLU} \circ \text{Linear} \circ (\text{MaxPool2D} \circ \text{ReLU} \circ \text{Conv2D})^{(4)}(\text{img}). \qquad (20)$$

We then standardize the output logits of $C_{\text{Attr}}$, for each attribute individually.

As for MorphoMNIST, we train one normalizing flow for each attribute. For this, we use the standardized logits from the classifiers rather than the original binary attributes from the data set. To model the non-Gaussian distributions, we employ the following flow architecture:

$$f_{\text{Attr}}(t, u_{\text{Attr}}) = \text{ScaleShift}\left(\left[\left(\text{QuadraticSpline}^{(10)} \circ \text{ConstScaleShift}\right)(u_{\text{Attr}}),\ \mathbf{x}_{\text{pa(Attr)}}\right]\right),$$

For the $\beta$-VAE with $\beta = 3$, we follow a slightly different approach as for B.3.2. Rather than concatenating the conditional variables $\mathbf{x}_{\text{pa}(i)}$ at the end of the encoder, we instead create an additional channel $\text{chan}_{\text{attr}}$ for each attribute attr that we concatenate to the RGB channels of the image. Specifically, we obtain the channel by broadcasting the continuous attribute value $x_{\text{Attr}}$ like

$$\text{ch}_{\text{Attr}} = \mathbf{1}_{128 \times 128} \cdot x_{\text{Attr}},$$

where we replace the (non-linear) neural network by a linear function for Bald, since the signal-to-noise ratio is low for this variable. The reason is that Beard is the only variable that cannot be modeled well as a linear function of its causal parents Age and Gender.

where $\mathbf{1}_{128 \times 128}$ is a matrix of dimensionality $128 \times 128$ that consists only of 1. We then feed $\tilde{\mathbf{x}} :=[x_R, x_G, x_B, \mathrm{ch}_{\mathrm{Beard}}, \mathrm{ch}_{\mathrm{Bald}}, \mathrm{ch}_{\mathrm{Gender}}, \mathrm{ch}_{\mathrm{Age}}] \in \mathbb{R}^{128 \times 128 \times 7}$ directly into the encoder with the following architecture (roughly inspired by Ghosh et al. (2020)):

$$f_{\mathrm{Img}}(\tilde{\mathbf{x}}) \approx e_{\mathrm{Img}}(\tilde{\mathbf{x}})$$
$$= \mathrm{Linear} \circ \mathrm{Pool2D} \circ (\mathrm{ReLU} \circ \mathrm{BatchNorm2D} \circ \mathrm{Conv2D})^{(6)}(\tilde{\mathbf{x}}),$$

where the final linear layer has output dimension $\mathtt{out} = 512$ and the transpose convolution layers (starting with parameters from the layer closest to the input) are parameterized by $\mathtt{out\_channels} = (128, 128, 128, 256, 512, 1024)$, $\mathtt{kernel\_size} = (3, 3, 3, 3, 3, 3)$, $\mathtt{stride} = (2, 2, 2, 2, 2, 2)$, $\mathtt{padding} = (1, 1, 1, 1, 1, 1)$. For the decoder, noting that $\mathbf{x}_{\mathrm{pa(Img)}} = [x_{\mathrm{Beard}}, x_{\mathrm{Bald}}, x_{\mathrm{Gender}}, x_{\mathrm{Age}}]$, we use

$$f_{\mathrm{Img}}^{-1}(\mathbf{x}_{\mathrm{pa(Img)}}, u_{\mathrm{Img}}) \approx d_{\mathrm{Img}}(\mathbf{x}_{\mathrm{pa(Img)}}, u_{\mathrm{Img}})$$
$$= \mathrm{TransConv2D} \circ (\mathrm{ReLU} \circ \mathrm{BatchNorm2D} \circ \mathrm{TransConv2D})^{(4)} \circ \mathrm{Linear}\left([\mathbf{x}_{\mathrm{pa(Img)}}, u_{\mathrm{Img}}]\right),$$

where the first linear layer maps to $\mathbb{R}^{4 \cdot 1024}$, which is then reshaped to a feature map in $\mathbb{R}^{2 \times 2 \times 1024}$. The consecutive transposed convolutional layers have the parameters $\mathtt{out\_channels} = (512, 256, 128, 128, 128)$, $\mathtt{kernel\_size} = (3, 3, 3, 3, 3)$, $\mathtt{stride} = (2, 2, 2, 2, 2)$, $\mathtt{padding} = (1, 1, 1, 1, 1)$.

## C  Ground Truth Structural Equations in Morpho-MNIST

The structural equation for thickness $T$ and intensity $I$ are given as

$$T \leftarrow 0.5 + U_T, \qquad\qquad\qquad U_T \sim \Gamma(10, 5)$$
$$I \leftarrow 191 \cdot \mathrm{Sigmoid}\left(0.5 \cdot U_I + 2 \cdot T - 5\right) + 64, \qquad U_I \sim \mathcal{N}(0, 1).$$

For details about how the MNIST images were modified as to change perceived thickness and intensity, we refer the reader to Pawlowski et al. (2020).

# D Additional Experiments

## D.1 Morpho-MNIST

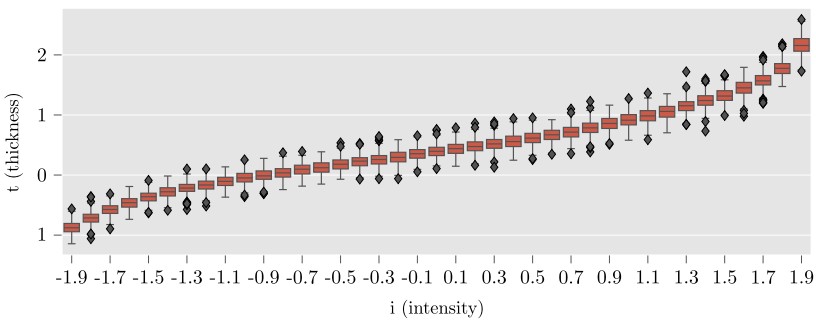

Figure 10: We run Fig. 4 **(b)** using stochastic DeepBC to sample from the distribution over counterfactual thickness values rather than just obtaining the mode. The box plots are generated from 400 samples per antecedent value.

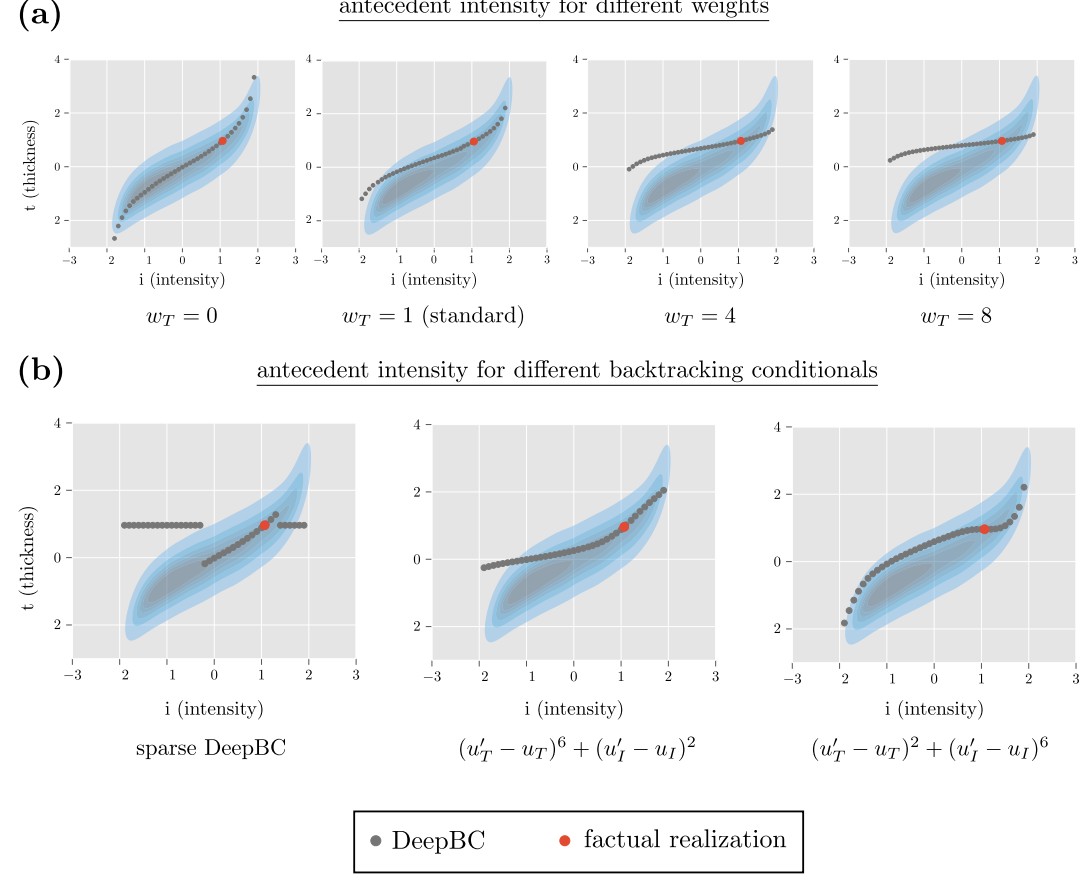

Figure 11: **(a)** We run Fig. 4 **(b)** multiple times, fixing $w_I = w_{\text{Img}} = 1$ and changing only $w_T$. We see that the backtracking solution approaches the interventional solution (see Fig. 4 **(b)**) as we increase $w_T$, thus preserving the value of thickness more as we increase weight. We note that the left-most plot ($w_T = 0$) corresponds to Fig. 4 **(c)**. **(b)** DeepBC can be extended to more general backtracking conditionals (corresponding distance functions plotted below each subplot) that lead to different solutions.

### D.2 CelebA

#### D.2.1 Additional Qualitative Experiments

Fig. 12 shows further plots related to CelebA (§ 4.2). Pannel **(a)** demonstrates two examples with multivariable antecedents. For the purpose of illustration, we also show single-variable antecedents for comparison. Pannel **(b)** showcases an example where the antecedent value $(x^*_{\text{Beard}})$ is solely absorbed by $u_{\text{Beard}}$, leaving all other high-level features unchanged, similar to an interventional counterfactual. This may be explained by the fact that many men are not bearded and thus this change does not need to be traced back to other variables. This is not possible if the noise in the predictor that "ought to be explained" is not modeled explicitly, such as in many counterfactual explanation methods, as shown on the right of pannel **(b)**. Specifically, the tabular explanation method can only generate a different prediction if it changes other variables, as can be seen by the changed gender in the counterfactual. Pannel **(c)** demonstrates the result of using a wrong graph structure, including the wrong graph employed, for the counterfactual demonstrated in Fig. 6. We also display the factual and sparse DeepBC result with the assumed graph structure (Yang et al., 2021b) for easier comparison. We indeed observe a large difference in the two counterfactuals.

#### D.2.2 Quantitative Experiments

In addition to the qualitative experiments in § 4.2 and App. D.2.1, we demonstrate quantitative experiments where we assess three metrics over a large sample of generated counterfactuals for the different methods. In addition to baselines 1 and 2 from the main text (§ 4.2), we assess a deep counterfactual explanation method (10). Specifically, we use an image regressor $(f_{\hat{Y}})$, together with an *unconditional* image auto-encoder $(f_X)$ to generate counterfactual explanations, as sketched in Section § 3.2. We do so for each attribute $(\hat{Y})$ separately. This corresponds in style to how Jacob et al. (2022); Rodríguez et al. (2021) obtain counterfactual explanations for images. This means that $f_X$ does not correspond to the reduced form of a causal model, and so $u_X = f_X^{-1}(x)$ corresponds to an unstructured embedding space for image data. For our experiments, we apply DeepBC in this unstructured latent space (where neither the attributes nor their causal relationships are modeled) and use classifiers to extract the high-level attributes from the counterfactual image. We also use DeepBC with a wrong graph to assess the effect of model missspecification on the result. The wrong graph used is shown in Fig. 12 **(c)**.

**Quantitative Evaluation Metrics.**   We evaluate three metrics: plausibility, observational closeness and causal compliance.[8] We define these as

$$
\begin{aligned}
\text{plausible}(\mathbf{x}^*) &:= \sum_{A \in \text{Attr}} - \log p\left(x^*_A \mid \mathbf{x}^*_{\text{pa}(A)}\right)/n, \\
\text{obs}(\mathbf{x}, \mathbf{x}^*) &:= \sum_{A \in \text{Attr}} m\left(x_A, x^*_A\right)/n, \\
\text{causal}(\mathbf{x}, \mathbf{x}^*) &:= \sum_{A \in \text{Attr}} m\left(f_A^{-1}(\mathbf{x}_{\text{pa}(A)}, x_A), f_A^{-1}(\mathbf{x}^*_{\text{pa}(A)}, x^*_A)\right)/n,
\end{aligned}
\tag{21}
$$

for $A \in \text{Attr} := \{\text{Age}, \text{Beard}, \text{Gender}, \text{Bald}\}$, $m$ denotes a distance function and $n = 4$. All of the metrics in (21) should be minimized. We restrict ourselves to the attributes for the computation of the metrics, because comparisons of images are generally problematic due to the high dimensionality. When comparing in terms of high-level attributes, this problem is alleviated and the metrics can be interpreted more easily. We use the squared distance $m(x, x') = (x - x')^2$ (SQU) and absolute distance $m(x, x') = |x - x'|$ (ABS).

The *plausible* metric measures the probability density of the generated counterfactual attributes, under the data distribution. In this sense, we can think of counterfactuals with high probability density (or low -log density) as being more plausible, i.e., being closer to the data manifold (see Karimi et al. (2022)). The *obs* metric measures the distance among attributes between factual realization and counterfactual, thereby evaluating closeness in the observation space (regardless of whether the causal laws are preserved).

---

[8]We note that counterfactuals cannot be validated, because ground truths do not exist. The purpose and quality of counterfactuals depend on the application domain and a universal metric does not exist.

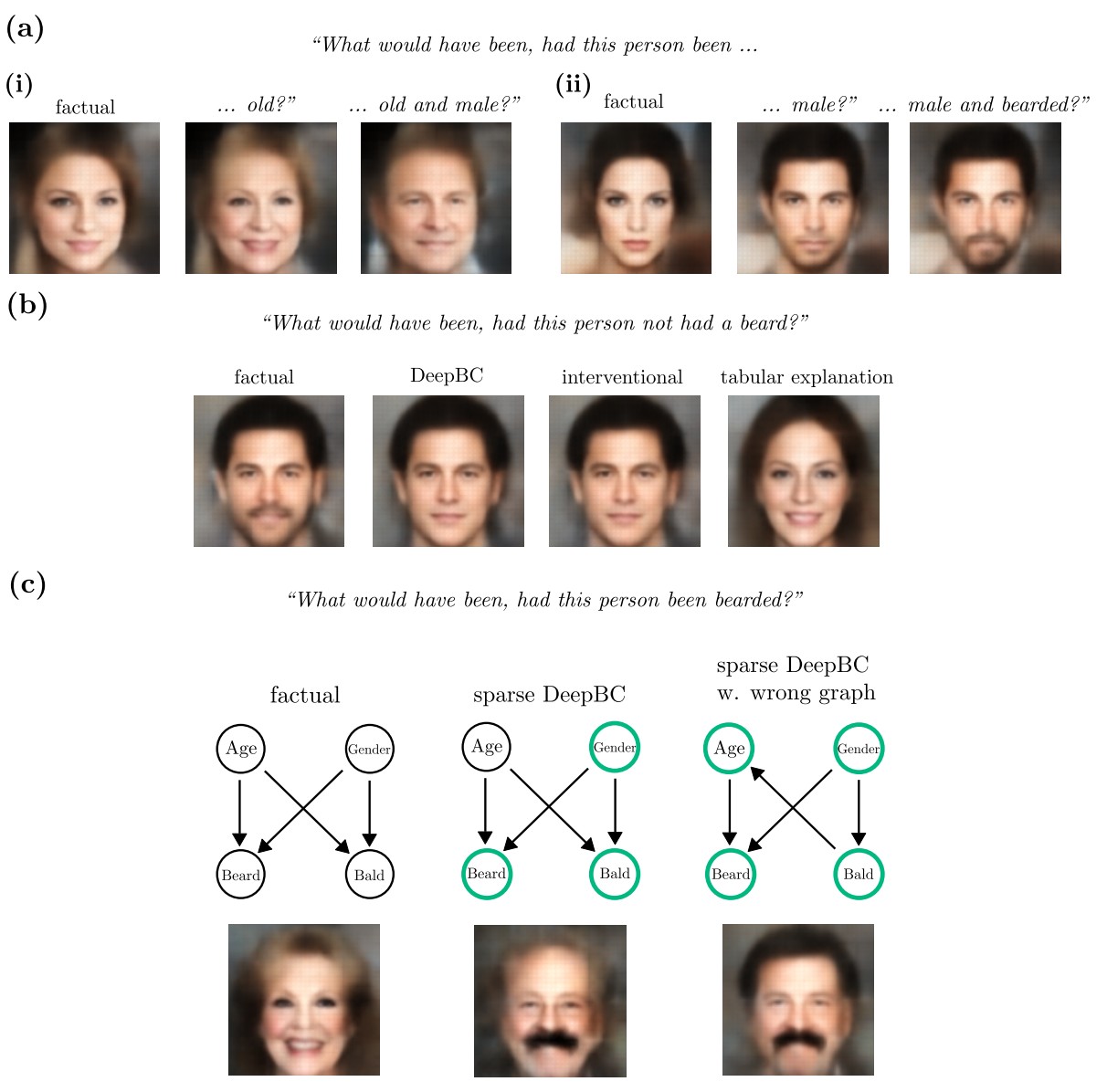

Figure 12: **Additional plots for CelebA. (a)** Two examples of multivariable DeepBC (**(i)** and **(ii)**). This means that not only one variable, but also multiple variables can be used as antecedent for DeepBC. **(b)** DeepBC takes into account non-deterministic relationships between variables: In this setting, the removed beard is traced back to $u_{\mathrm{Beard}}$ rather than other variables. The result is highly similar to the interventional example (plotted for comparison) and very different to the tabular counterfactual explanation method, which must change either one of the other attributes (here gender) in order to achieve a different model prediction. **(c)** Applying sparse DeepBC with a wrong graph produces a different result compared to using the correct graph. Specifically, we observe that all factual variables change their value due to the far-reaching downstream effects of the gender variable in the wrong graph.

This is the metric that has been optimized in the original work on counterfactual explanations (Wachter et al., 2017). The *causal* metric measures the difference in exogenous variables between realization and counterfactual, which can be interpreted as the degree of preservation of the causal mechanisms: If changes in latent variables exceed what is minimally necessary under the assumption of retained mechanisms, it suggests that the counterfactual does not effectively preserve the causal mechanisms. Since we do not have access to ground truth structural equations $(f_1, f_2, ..., f_n)$ in CelebA, we use the ones that were trained on

Table 1: **Quantitative Evaluations for CelebA.** Three considered metrics ± standard deviation of the different baselines for 500 iterations. Lower values are better, best per metric is highlighted in bold. *CE* stands for *counterfactual explanation* (see § 2.4). The two distance functions assessed ($m$) are SQU for squared distance, and ABS for absolute distance.

| Metric | m | Tabular CE | Interventional | Deep CE | Wrong graph | DeepBC |
|---|---|---|---|---|---|---|
| obs | SQU | $1.778 \pm 1.883$ | $\mathbf{0.513 \pm 0.892}$ | $0.852 \pm 0.691$ | $0.690 \pm 1.076$ | $0.611 \pm 0.985$ |
| | ABS | $0.879 \pm 0.575$ | $\mathbf{0.345 \pm 0.317}$ | $0.720 \pm 0.305$ | $0.545 \pm 0.451$ | $0.451 \pm 0.372$ |
| causal | SQU | $2.304 \pm 2.620$ | $0.674 \pm 1.140$ | $1.286 \pm 0.983$ | $0.504 \pm 0.893$ | $\mathbf{0.498 \pm 0.930}$ |
| | ABS | $1.080 \pm 0.748$ | $0.314 \pm 0.265$ | $0.887 \pm 0.360$ | $0.400 \pm 0.327$ | $\mathbf{0.310 \pm 0.270}$ |
| plausible | | $0.984 \pm 1.424$ | $0.374 \pm 0.439$ | $\mathbf{0.153 \pm 0.281}$ | $0.311 \pm 0.373$ | $0.301 \pm 0.358$ |

the data set. We furthermore note that incorporating the antecedent variable into the loss is not an issue either, because it is fixed for all methods. For the deep non-causal explanation method, we only obtain the counterfactual image, without explicit access to the attribute variables. In order to extract those, we use the (standardized) logits of classifiers that were trained to predict the attributes from the image.

We obtain the numbers in table Tab. 1 as follows: We sample a factual data point $\mathbf{x} = \mathbf{F}(\mathbf{u}), \mathbf{u} \sim \mathcal{N}(\mathbf{0}, \mathbf{I})$. Then, we sample an attribute uniformly, i.e.

$$a \sim \mathcal{U}(\{\text{age, gender, beard, bald}\})$$

and construct the corresponding antecedent as

$$x_\text{a}^* \sim \mathcal{N}(0, 1).$$

We then compute the counterfactual $\mathbf{x}^*$ to evaluate all three loss functions (21). This process is repeated 500 times. The final reported scores are the arithmetic means over the individual metrics, including $\pm 1$std.

**Results.** We quantitatively assess multiple properties of sparse mode DeepBC in comparison to the baselines in Tab. 1. While the deep counterfactual explanation method (10) achieves the best result on the *plausible* score, the interventional approach preserves the attribute values the best (obs score). This is likely due to the fact that counterfactuals on leaf nodes of the causal graph in Fig. 6 do not change the values of their parents at all, in contrast to the other methods. Finally, as expected, DeepBC obtains the best score on the *causal* metric, independent of distance function $m$. This can be explained due to the fact that DeepBC always respects downstream effects (in comparison to other methods), as exemplified in Fig. 6.

