# OpenReview forum: "Deep Backtracking Counterfactuals for Causally Compliant Explanations"
_TMLR — Accepted by TMLR_

### Review · Reviewer_Hh2N · 2024-04-02

**Summary Of Contributions:**

The paper proposes a MCMC-based algorithm in the structured latent space of what is referred to as a causal model, based on the "overdamped Langevin dynamics (Parisi, 1981)". The paper also proposes a simplified method that provides a single counterfactual based on a constrained optimization problem.

**Audience:**

Yes

**Claims And Evidence:**

Yes

**Requested Changes:**

In terms of the algorithm, the key building blocks are hidden in the formalization. In particular, the following elements weren't clear to me:
- How is the learning of U = F−1(X) done? It might be interesting to state what is given in the experiments. For instance in context of the celebA dataset, does the algorithm have access to all the labels (e.g. gender, baldness, etc.)?
- How is the methods described in 3.3.1 different than e.g. diffusion models? And is this a key part of the contribution (only references are from 1981 and 2013)?
- In Fig 9 for the plot $\lambda=10^6$, why is there some periodicity in the blue loss?
- In the experiments section, can you clarify what experiments are related to "Langevin Monte Carlo sampling" and the ones related to "constrained optimization"?

**Strengths And Weaknesses:**

Weaknesses:
- Some elements/hypotheses in the paper are not always clearly related to the literature in particular in the methodology section (see changes/questions below). This makes the contributions sometimes a bit hard to pinpoint.

Strengths:
- The paper tackles an interesting problem and shows interesting results
- Anonymized source code is available

Disclaimer: This paper is not in my core expertise.

---

> ### Author Response · Authors · 2024-05-01
>
> We appreciate that you find both the problem under consideration and the results interesting. We furthermore thank you for carefully assessing the paper and providing helpful suggestions. We proceed to address each one of your requested changes:
>
> > How is the learning of U = F-1(X) done? It might be interesting to state what is given in the experiments. For instance in context of the celebA dataset, does the algorithm have access to all the labels (e.g. gender, baldness, etc.)?
>
> We do explicitly mention observability of all $X_i$ in section 2.1 and in the discussion section (Section 6), in the paragraph *Identifiability of the Reduced Form*. We do not mention it explicitly in the CelebA section (Section 4.2). We thus decided to extend description of the experimental setup with the following sentence:
>
> *The images have a resolution of 128×128 and are annotated with binary attributes {Age,
> Gender, Beard, Bald}. Both the image and the attributes correspond to the observable variables.*
>
> We extended the experimental setup of MorphoMNIST analagously. We hope that this clarification is helpful.
>
> > How is the methods described in 3.3.1 different than e.g. diffusion models? And is this a key part of the contribution (only references are from 1981 and 2013)?
>
> The method described in Section 3.3.1 is distinct from diffusion probabilistic models. In our setting, the gradient of the energy is not time-dependent and is not directly trained on data. Instead, our energy function can be derived from the backtracking conditional and the deep SCM. We consider this technique for sampling counterfactuals to be a contribution, as stated in the first bullet point of our contributions in Section 1.
>
> > In Fig 9 for the plot $\lambda = 10^6$, why is there some periodicity in the blue loss?
>
> We believe that the periodicity stems from small eigenvalues of $\lambda^{-1}\mathbf{W} + \mathbf{J}_S^{\top} \mathbf{J}_S$, as written in Section B.1:
>
> *The plot for $\lambda = 10^6$ shows that the linearization method can lead to oscillations if $\lambda$ is chosen too large, which likely stems from small eigenvalues of $\lambda^{-1}\mathbf{W} + \mathbf{J}_S^{\top} \mathbf{J}_S$ (we not that $\mathbf{J}_S^{\top} \mathbf{J}_S$ is low-rank) that give rise to numerical issues*
>
> We would be happy to provide further clarification if needed.
>
> >In the experiments section, can you clarify what experiments are related to "Langevin Monte Carlo sampling" and the ones related to "constrained optimization"?
>
> We do clarify our terminology in this regard in the experiments section (Section 4):
>
> *As showing a single example per counterfactual is more illustrative than sampling multiple examples, we mainly focus on mode DeepBC and refer to this variant when using the term DeepBC from this point on.*
>
> In response to your suggestion, we added a sentence to clarify this terminology further:
>
> *When referring to stochastic DeepBC, we always explicitly write stochastic DeepBC.*

---

### Review · Reviewer_2Fj7 · 2024-04-05

**Summary Of Contributions:**

This paper presents Deep Backtracking Counterfactuals (DeepBC), a practical implementation of the recently proposed backtracking counterfactuals idea [von Kügelgen et al., CLeaR 2023]. Two variants are proposed, stochastic DeepBC and mode DeepBC, with practical algorithmic implementations. Experiments are conducted on two datasets, Morpho-MNIST and CelebA, where the method is compared against interventional counterfactuals as well as several ablated versions as baselines.

**Audience:**

Yes

**Broader Impact Concerns:**

This is a generic ML paper, no particular concern

**Claims And Evidence:**

No

**Requested Changes:**

Critical:
 - clarify what the non-causal methods actually do, and what are their expected limitations compared to DeepBC
 - either report other, non-biased metrics, or motivate why these metrics make sense although they are biased, or simply remove the quantitative experiments from the paper

Strengthen the paper:
 - discuss how backtracking counterfactuals differ from regular, observational counterfactuals (comment p4: Backtracking counterfactuals)
 - better motivate how DeepBC provides causally compliant explanations, although the method gives the same result regardless of causal directions in the graph (as long as no new independence is introduced, i.e. a fully connected graph is fine), or prove me wrong if I am mistaken (comment p9: Causal compliance)

**Strengths And Weaknesses:**

Strengths:
 - the paper is well-written and easy to follow, the derivations clear
 - the relation to existing literature (counterfactual explanations) I found was interesting
 - the proposed method is simple

Weaknesses:
 - it is not clear to me what the non-causal baselines (tabular and deep) are actually doing.
 - I found the quantitative metrics and results flaky and questionable. It is not clear what should be expected of each method, and it is very questionable that the evaluation involves the model learned by DeepBC.
 - the statement of the title, that backtracking counterfactuals allow for causally compliant explanations, is not well supported in the paper. See my detailed comments (p4: Backtracking counterfactuals, p9: Causal compliance).

Detailed comments:

 - p3: the causal parents of Xi as specified by G -> A detail is missing at this point. Does G encode the causal relationships between endogenous variables only (X)? Or between all variables (X,U)? In other words, do you allow exogenous variables in U to be causally related? (I can find the answer later, but this should be stated here)

 - p3: compted -> typo

 - p4: Backtracking counterfactuals -> Consider yet another latent variable, say $\bar{U}$, which would be a parent of both $U$ and $U'$. Then, one can express the backtracking conditional as the marginalized distribution $p(u'|u)=\sum_{\bar{u}} p(\bar{u}|u)p(u'|\bar{u})$ without loss of generality, and backtracking counterfactuals can be re-written as regular counterfactuals where $X$, $\bar{U}$ are respectively the exogenous and endogenous variables. If so, aren't backtracking counterfactuals simply a parametric instantiation (causal mechanisms expressed with intermediate exogenous variables) of the more general (non-deterministic) counterfactual framework? Is there something fundamental in backtracking counterfactuals that regular (observational) counterfactuals cannot express?

 - p4: p(u'|u) -> why this notation, instead of p(u*|u) ? Is there a difference between u' and u*?

 - p5: $p^B(u'|u)$ -> Neither the latent distribution p(u|x) nor the backtracking conditional p(u'|u) are identifiable from observational datasets. Is this limitation discussed at all somewhere?

 - p7: $d_i(u'_i,u_i)$ -> Why this particular choice of a distance function? It is the first time this distance function is explicitly defined. Can DeepBC account for arbitrary distance functions? Is this particular choice of distance function an assumption of DeepBC? This should be clarified earlier in the manuscript.

 - p9: Causal compliance -> I find this point a bit misleading. It seems that DeepBC generates observational counterfactuals, and hence will produce the same output even if the causal structure is miss-specified, as long as it does not violate the dependencies in the data distribution. For example, in Figure 4, I do believe that DeepBC will produce the same result with the causal arrow reversed. And in Figure 6, DeepBC should produce the same results with any fully connected causal graph. Hence I fail to see how DeepBC produces causally compliant explanations, as it does not exploit causality at all (only the independence model). This point should at least be clarified in the Discussions section.

 - p10: We use here $d_i(u'_i, u_i)$ -> It would be nice to recall what are the exogenous variables $U_i$ here. Are there only three, for T, I and the image?

 - p12: Tabular non-causal explanation -> I do not understand this baseline. How is the image reconstructed from the attributes? What is the distance function here?

 - p12: Deep non-causal explanation -> Likewise, I have a hard time understanding this baseline. What is the difference with baseline 1? Is the distance function applied here only on $u_x$, or also on the attributes? Do baselines 1 and 3 correspond to the counterfactual explanations method described in 2.4?

 - p13: Table 1 -> The standard deviations in this table are extremely large, how is that? Moreover, the numbers reported in this table are I think very questionable, given that 1) the objective of each method is not the same and 2) both the causal and plausible metrics are biased because they rely on the normalizing flows learned by DeepBC during training. Moreover, the so-called "causal" metric is exactly the objective minimized by DeepBC (6)... This is like proposing a problem after the solution has been defined. I find Table 1 extremely confusing, and I am not sure what information I can extract from it. Why is a deep non-causal method better at producing plausible explanations? This is not even discussed in the text. I would suggest to either better motivate the experimental setup and discuss the numbers, or remove that table.

 - p15 Discussion -> I was waiting for a discussion of the limitations of this work, in particular on the fact that neither the latent distribution p(u|x) nor the backtracking conditional p(u'|u) are in general identifiable from observational data. I am glad to find some mention of it here, although it could be made more explicit. It is perfectly fine to present a method with limitations as long as these are acknowledged and discussed. For the reader's sake, I suggest to rename this section more explicitly "Limitations".

---

> ### Author Response · Authors · 2024-05-01
>
> We appreciate your thorough assessment of our manuscript and are pleased that you find it well-written and interesting. We found your comments and questions very helpful and have taken them into account for our revised manuscript. This strengthened the clarity of our method and the demonstration of its causal compliance.
>
> We first address your critical requested changes:
>
> > clarify what the non-causal methods actually do, and what are their expected limitations compared to DeepBC
>
> To address this, we extended the paragraph introducing the tabular explanation baseline and introduced the deep non-causal explanation method earlier in Section 3.2. Additionally, we conducted additional experiments on Morpho-MNIST to demonstrate the limitations of non-causal methods. These are discussed in the last paragraph of Section 4.1 and in the last paragraph of the discussion section. We furthermore renamed the last section to *Limitations of Non-Causal Explanation Methods.* The new Figure 4 **(b)** shows multiple examples where using the wrong graph (which also yields a fully connected graph) leads to causally incompliant results. The true causal graph, however, is generally impossible to identify for non-causal methods, which directly implies the limitation of non-causal methods.
>
> > either report other, non-biased metrics, or motivate why these metrics make sense although they are biased, or simply remove the quantitative experiments from the paper.
>
> We have moved the numerical experiments to the appendix and are open to conducting further changes (also removing the quantitative metrics altogether).
>
> We also address your detailed comments one-by-one:
>
> - Causal relation between exogenous variables: We extended the paragraph with a footnote to clarify that exogenous variables are not causally related to each other.
>
> - Typo: This issue has been fixed in our revised manuscript.
>
> - Expressivity of observational counterfactuals: Figure 6 demonstrates an example where observational counterfactuals (tabular counterfactual explanations that measures distance in $X$ and not in $U$) violate a causal relationship. Specifically, the generated observational counterfactual does not automatically take into account the causal downstream effect (being male and old leads to baldness), which happens because distance is not measured in $U$, but in $X$. We decided to provide additional intuition for "causal compliance" at the end of Section 3.2.
>
> - Notation for backtracking conditional: We use the notation to avoid incorrect equation formulation, which would otherwise read $\delta_{\mathbf{x}_S^\ast} (\mathbf{x}_S^\ast)$, where $\mathbf{x}_S^\ast$ refers to the antecedent, which does not make sense.
>
> - Identifiablity of reduced form and backtracking conditional: The problem of identifiability of the reduced form is discussed in Section 6 and we have added a footnote to Section 3.1.1 to emphasize that the backtracking conditional is not learned from data.
>
> - Choice of distance function: We decided to conduct further experiments with different distance functions and added the results in the new Figure 4 and Figure 11.
>
> - Causal compliance: We decided to further elaborate on the concept of "causal compliance" at the end of Section 2.3 and provided additional experiments on Morpho-MNIST to illustrate the dependence on the graph, beyond violation of conditional independencies, as shown in Figure 4. Specifically, we see that using the wrong graph (despite it being fully connected) leads to causally incompliant results.
>
> - Clarification of exogenous variables: We added clarification regarding the notation $u_i$ in the paragraph.
>
> - Clarification of the tabular baseline: In the revised manuscript, we explicitly state that we employ sparse DeepBC on the observed attributes and elaborate on how we generate the image.
>
> - Clarification of the deep non-causal baseline: We removed the paragraph introducing deep counterfactual explanations in the experiments section to avoid confusion. Instead, we now introduce it earlier, since the deep counterfactual explanation method corresponds to equation (10).
>
> - CelebA experiments: The numerical results from the CelebA section have been moved to the appendix, and the causal compliance of our method is demonstrated through additional experiments on Morpho-MNIST.
>
> - Renaming the discussion section: We renamed the section to "Discussion & Limitations" to reflect the broader scope of the discussion.

---

### Review · Reviewer_PkgY · 2024-05-06

**Summary Of Contributions:**

This work studies and introduces causal inference using backtracking counterfactuals and deep invertible generative neural networks. In contrast to interventional causal calculus, the backtracking approach considers an unmodified causal structure and examines differences in a latent factor factor model. The authors parameterize this difference using a simple exponentiation, and consider invertible causal generative deep models. Experiments are performed which evaluate the performance of the proposed algorithm under misspecification of the underlying causal dag, and on counterfactual image generation.

**Audience:**

Yes

**Broader Impact Concerns:**

I do not believe that this paper requires adding a Broader Impact Statement.

**Claims And Evidence:**

Yes

**Requested Changes:**

It would be great if the authors could:
1. Provide a set of experiments / theoretical results / more extensive discussion which describes what happens under model misspecification.
2. Make precise the implications of invertibility in terms of the space of dependence functions that can be modeled.
3. Provide some sort of error bound / discussion around worst case behavior here in terms of error estimating the latent factors translating to errors in the inferred counterfactual.

**Strengths And Weaknesses:**

Strengths:
1. The use of the idea of backtracking counterfactuals within the machine learning community is still fresh and provides an interesting perspective on counterfactual inference. This paper does a nice job of taking those ideas an applying it to a more applied and modern context by connecting it to deep generative models.
2. The provided experimental results show strong empirical performance.
3. The authors do a nice job of describing the task, and necessary background for the paper. The discussion of interventional versus backtracking counterfactuals was particularly well done and makes for a very accessible introduction. Additionally, I felt the overall organization and writing was quite strong.

Weaknesses:
The work seems to hinge on a few key aspects:
 (1) correct specification of the underlying DAG. In my view the authors do a nice job of addressing this with their experiments.
 (2) The correct specification of the latent factors, it's unclear what the implications are here for misspecification of the latent factors. What happens if we are wrong about the cardinality of them, for example? I assume this would have a significant effect on the entailed distance metric. In addition, it's not clear when we should assume relatively low rank for these latent factors.
 (3) The invertibility of the underlying generative model. It's not entirely clear to me what the downstream ramifications are of enforcing invertibility. It would seem to me that this would limit the space of functions that can describe dependence between any two variables. Is it possible to make this precise?
 (4) It also isn't entirely clear to me how one decides whether a model has sufficiently captured the aspects of the generating system such that we can trust the downstream counterfactuals. Is it possible to make this precise?

---

> ### Author Response · Authors · 2024-05-19
>
> We are pleased to receive your positive feedback on our manuscript and appreciate your insightful questions. We have carefully considered each of your suggestions and made the following revisions:
>
> > Provide a set of experiments / theoretical results / more extensive discussion which describes what happens under model misspecification.
>
> We have included an additional paragraph titled *Model Misspecification* in our discussion section. Here, we emphasize that DeepBC may not produce correct backtracking counterfactuals under model misspecification. Additionally, we reference literature that discusses choice of the latent dimensionality for variational autoencoders.
>
> > Make precise the implications of invertibility in terms of the space of dependence functions that can be modeled.
>
> In the *Model Misspecification* paragraph mentioned above, we also discuss the common assumption of invertible mechanisms (also known as bijective generation mechanisms), which we also make in our approach. We identify exploring non-invertible generative models as a promising avenue for future work to address this limitation.
>
> > Provide some sort of error bound / discussion around worst case behavior here in terms of error estimating the latent factors translating to errors in the inferred counterfactual.
>
> Finite sample size analysis is beyond the scope of our current work. However, we do believe that results from statistical learning theory can likely be extended to derive error bounds on the generated counterfactuals.

---

### Author Response · Authors · 2024-05-01
**General Response (updated)**

We sincerely thank the reviewers for dedicating their time and effort to evaluate our manuscript. We are delighted that the reviewers recognize the importance of the problem we address ("The paper tackles an interesting problem and shows interesting results" - Hh2N) and appreciate its connection to existing literature ("the relation to existing literature (counterfactual explanations) I found was interesting" - 2Fj7; "[...] provides an interesting perspective on counterfactual inference" - PkgY). Moreover, we are grateful for the positive remarks on the clarity and simplicity of our presentation ("the paper is well-written and easy to follow, the derivations clear" - 2Fj7; "Additionally, I felt the overall organization and writing was quite strong." - PkgY; "the proposed method is simple" - 2Fj7).

We highly value the constructive feedback provided by the reviewers, and we have incorporated their suggestions into the revised manuscript. We have addressed each comment and believe that these revisions significantly strengthened our paper.

We highlight the most salient changes to our manuscript:

1. We added additional experiments to the Morpho-MNIST section (Section 4.1 and Figure 4) to demonstrate the importance of employing the true causal graph, beyond the encoded conditional independences. We also added an additional section to the end of Section 2.3 that intuitively describes what we mean by causal compliance using the concrete example of Figure 2.

2. We moved the quantitative experiments of celebA to the Appendix (Now in Appendix D.2.2).

3. We extended the "Discussion \& Limitations" section by one more paragraph called "Model Misspecification" that discusses the limitations of our method with respect to model misspecification and suggests possible remedies.

All changes in the updated manuscript are highlighted in orange.

---

### Decision · Action_Editor_zov2 · 2024-07-16

**Recommendation:** Accept as is

**Comment:**

The reviewers generally found the work to be insightful and contributing to the state of the art for counterfactual reasoning.  They raised concerns about the assumptions and conditions needed for the techniques to work and the authors added suitable explanations to clearly outline the limits of the proposed techniques.  One reviewer further requested that bounds be derived in situations where the necessary conditions are not met or unverifiable, but this is beyond the scope of this work.  The proposed techniques do represent an important advance for backtracking counterfactuals in the context of modern deep learning models.  Furthermore, the paper provides important insights about the differences between non-causal counterfactual methods, interventional counterfactuals and backtracking counterfactuals.

**Audience:**

Anyone interested in counterfactual estimation.

**Claims And Evidence:**

The paper proposes new techniques to estimate backtracking counterfactuals (in contrast to interventional counterfactuals).  The paper clearly outlines the assumptions and conditions needed for the techniques to work.  While the required conditions may not always hold or be verifiable in practice, the contributed techniques do represent an important advance with respect to previous work.  Experiments demonstrate the effectiveness of proposed techniques.